# Nef-Net v2: Adapting Electrocardio Panorama in the Wild

**Zehui Zhan**[1,2,*]**, Yaojun Hu**[3,*]**, Jiajing Zhang**[4]**, Wanchen Lian**[1]**, Wanqing Wu**[2,†]**, Jintai Chen**[1,5,†]

[1] Information Hub, Hong Kong University of Science and Technology (Guangzhou), Guangzhou, 511458, China

[2] School of Biomedical Engineering, Sun Yat-sen University, Shenzhen, 518107, China

[3] College of Computer Science and Technology, Zhejiang University, Hangzhou, 310012, China

[4] Department of Electrical and Electronic Engineering, University of Hong Kong, Hong Kong, 999077, China

[5] The Hong Kong University of Science and Technology, Hong Kong, 999077, China

*: Co-first authors; †: Corresponding authors

## Abstract

Conventional multi-lead electrocardiogram (ECG) systems capture cardiac signals from a fixed set of anatomical viewpoints defined by lead placement. However, certain cardiac conditions (*e.g.*, Brugada syndrome) require additional, non-standard viewpoints to reveal diagnostically critical patterns that may be absent in standard leads. To systematically overcome this limitation, Nef-Net was recently introduced to reconstruct a continuous electrocardiac field, enabling virtual observation of ECG signals from arbitrary views (termed *Electrocardio Panorama*). Despite its promise, Nef-Net operates under idealized assumptions and faces in-the-wild challenges, such as long-duration ECG modeling, robustness to device-specific signal artifacts, and suboptimal lead placement calibration. This paper presents Nef-Net v2, an enhanced framework for realistic panoramic ECG synthesis that supports arbitrary-length signal synthesis from any desired view, generalizes across ECG devices, and compensates for operator-induced deviations in electrode placement. These capabilities are enabled by a newly designed model architecture that performs direct view transformation, incorporating a workflow comprising offline pretraining, device calibration tuning steps as well as an on-the-fly calibration step for patient-specific adaptation. To rigorously evaluate panoramic ECG synthesis, we construct a new *Electrocardio Panorama* benchmark, called Panobench, comprising 4470 recordings with **48** views per subject, capturing the full spatial variability of cardiac electrical activity. Experimental results show that Nef-Net v2 delivers substantial improvements over Nef-Net, yielding an increase of around 6 dB in PSNR in real-world settings. Our code is publicly available at https://github.com/HKUSTGZ-ML4Health-Lab/NEFNET-v2.

## 1 Introduction

Cardiovascular diseases remain the leading cause of morbidity and mortality worldwide (Gaziano et al., 2006), claiming tens of millions of lives each year and imposing profound disability burdens that underscore an urgent clinical imperative (Tsao et al., 2023). Among diagnostic modalities, electrocardiogram (ECG) has established itself as indispensable, providing a non-invasive, cost-effective approach that offers immediate insights into the complex dynamics of cardiac electrical activity (van't Hof et al., 1997).

The number of ECG observation viewpoints directly correlates with both practical complexity and the comprehensiveness of cardiac condition understanding (Kligfield et al., 2007). The standard 12-lead ECG, which is widely used in clinical practice, is generally considered a practical compromise between acquisition cost and clinical utility (Holter, 1961). The electrode placement for the standard 12-lead ECG is illustrated in Figure 1(a). However, this conventional setup is still insufficient for detecting certain cardiac pathologies with specialized localization patterns, and may require additional, non-standard viewpoints. For instance, posterior myocardial infarction often requires additional

Table 1: A comparison of the synthesis performance between Nef-Net and NEF-NET V2, following an identical patient-level testing protocol.

| Method | ECG Length | Device | Lead Placement | Synthesis: PSNR($\uparrow$) |
|--------|------------|--------|----------------|-----------------------------|
| Nef-Net | Heartbeat | Restricted | High-Precision | 24.10-28.01 |
| NEF-NET V2 | Continuous | Agnostic | In-the-wild | 32.07(7.97$\uparrow$)-34.82(6.81$\uparrow$) |

posterior leads (V7-V9)[1] for definitive diagnosis (Van Gorselen et al., 2007). Brugada syndrome detection requires ECG acquisition from additional viewpoints (at 2nd/3rd ICS)[2], as these viewpoints uniquely capture the pathological signals caused by right ventricular outflow tract depolarization abnormalities (Berne & Brugada, 2012).

To address the trade-off between information richness and view availability in ECGs and to advance the clinical utility of panoramic ECG analysis, Chen et al. (2021) introduced *Electrocardio Panorama*, which constructs an implicit neural representation, enabling the generation of ECG signals from arbitrary viewpoints in real time. While this pioneering approach lays an important foundation, Nef-Net nonetheless shows several limitations that constrain its adoption:

**(1) Heartbeat-level modeling.** Nef-Net is confined to single-heartbeat ECG reconstruction, limiting clinical utility since continuous monitoring requires long-duration signals to capture inter-beat dynamics and arrhythmia patterns. **(2) Underuse of view-specific features.** Nef-Net constructs its electrocardio field representation by simply averaging the features from different viewpoints, thereby neglecting the varying relevance of each input view to the target view. This aggregation strategy blends information from all views uniformly, which not only often results in overly coarse reconstructions but also introduces unnecessary architectural complexity. This degradation is further exacerbated when only few views are available for supervision. **(3) Neglect of practical deployment factors.** Training assumes idealized data, and Nef-Net overlooks two real-world challenges: inter-device shifts (*e.g.*, variations in sensor characteristics and signal-processing pipelines across ECG devices) and inter-subject differences (*e.g.*, electrode placement offsets introduced by clinical staff). **(4) Limited validation of panoramic ECG.** Due to the unavailability of dense-view datasets, evaluation has been restricted to 12-lead ECGs under narrow angular settings, which is insufficient to assess the model's generalizability in reconstructing the global cardiac field. *We make the following contributions to address these limitations:*

(A) **Geometric View Transformer (GeoVT).** We introduce a geometry-aware cross-attention architecture that explicitly models spatial relationships between query and recorded ECG views, selectively extracting the most relevant features for direct query-view transformation. This approach overcomes the feature-averaging limitation in Nef-Net, enabling NEF-NET V2 to achieve superior synthesis performance while simplifying the modeling pipeline and reducing the overall parameter count.

(B) **A New Paradigm for Model Development.** We propose a unified three-stage pipeline for developing and deploying NEF-NET V2 in real-world Electrocardio Panorama applications. In the initial *Any-Pairs Pretraining* stage, the model acquires robust cross-view transformations under controlled laboratory conditions. Then, *Device Calibration* stage addresses feature distribution shifts arising from heterogeneity across ECG devices in clinical environments. Finally, the *On-the-fly Calibration* stage enables rapid, geometry-aware adaptation at the level of individual examinations, compensating for variations in electrode placement.

(C) **A Novel Panoramic ECG Dataset and Superior Performance.** To enable comprehensive benchmarking of Electrocardio Panorama synthesis, we curated *Panobench* (lead positions shown in Figure 1(c)), the first **48**-lead ECG dataset with precisely CT-measured spherical coordinates for each view, with inter-subject angular variance quantified across 4,470 recordings. Experimental results show that NEF-NET V2 accurately synthesizes novel ECG views, offering a promising avenue toward more comprehensive clinical ECG assessment.

---

[1] V7–V9 are ECG leads, beyond the standard 12-lead system, placed on the back to record activity from the posterior heart wall.

[2] The 2nd/3rd ICS are specific anatomical locations on the chest.

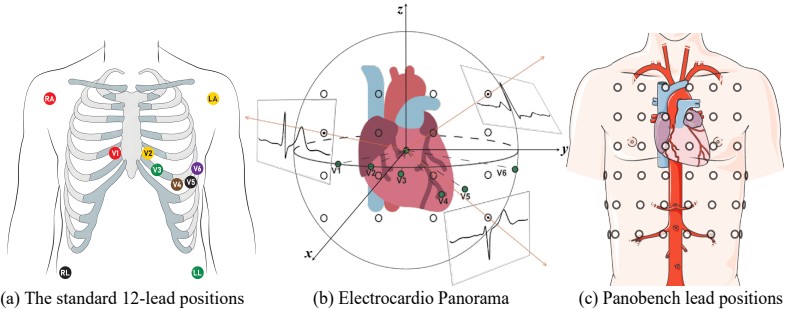

(a) The standard 12-lead positions    (b) Electrocardio Panorama    (c) Panobench lead positions

Figure 1: (a) Standard 12-lead electrode placement for conventional ECG recording. (b) *Electrocardio Panorama* enables any user-desired virtual ECG viewpoints for comprehensive visualization of cardiac electrical activity. (c) The Panobench benchmark, encompassing **48** distinct ECG viewpoints for each case, enables rigorous evaluation of Electrocardio Panorama generation models.

## 2    BACKGROUND AND RELATED WORK

### 2.1    ELECTROCARDIOGRAM (ECG)

ECG recordings are time-series signals of cardiac electrical activity. Each cardiac cycle can be decomposed into six non-overlapping deflections: the P wave, PR segment, QRS complex, ST segment, T wave, and TP segment. The standard 12-lead ECG protocol (lead positions shown in Figure 1(a)) is widely used for cardiovascular screening and typically captures 10-second recordings from 6 limb leads (I, II, III, aVR, aVL and aVF) and 6 chest leads (V1-V6) (Kligfield et al., 2007). Each lead acts as a distinct sensor, providing spatially distinct views of cardiac electrical activity, analogous to multi-view camera systems in computer vision. A more detailed introduction to ECGs is provided in Appendix B.

### 2.2    ECG VIEW RECONSTRUCTION AND SYNTHESIS

ECG view reconstruction is essential for recovering missing leads and enabling comprehensive cardiac assessment. Early methods relied on linear transformations (Nelwan et al., 2004), assuming predominantly linear relationships across leads, which fails to capture the inherently nonlinear cardiac dynamics (McCulloch et al., 1998). To address this limitation, nonlinear approaches have been developed, including recurrent neural networks (RNNs) (Hernandez-Matamoros et al., 2020), long short-term memory networks (LSTMs) (Kapfo et al., 2022), convolutional neural networks (CNNs) (Gundlapalle & Acharyya, 2022; Chen et al., 2024; Lence et al., 2025), and conditional generative adversarial networks (CGANs) (Seo et al., 2022; Golany & Radinsky, 2019; Joo et al., 2023), which better model complex inter-lead relationships. Yet, these methods are limited to reconstructing predefined, known views and cannot generate novel views that may be clinically valuable. Chen et al. (2021) first introduced the concept of *Electrocardio Panorama*, which enables the synthesis of any unseen views conditioned on viewing angles. While this represents a significant conceptual advance, the approach remains unsuitable for real-world clinical applications due to its neglect of real-world challenges like operational offsets and device inconsistencies. Our method addresses these constraints and provides a more robust solution for panoramic ECG observation.

## 3    METHODOLOGY

### 3.1    ARCHITECTURE

The key idea of NEF-NET V2 is to formulate ECG view synthesis as a direct view-to-view transformation problem. This is a pairwise deterministic mapping: the model converts the observed lead signals into the target lead through a single-step transformation, without modeling any shared geometric prior (*e.g.*, the *electrocardio field representation*) as Nef-Net (Chen et al., 2021). NEF-NET V2 incorporates three core components: **Angle Embedding**, **View Encoder**, and **Geometric View Transformer (GeoVT)**, as illustrated in Fig. 2. Formally, let $X = \{x_1, \cdots, x_l\}$ with each $x_i \in \mathbb{R}^{1 \times t}$ denote $l$ ECG signals recorded from distinct viewing angles.

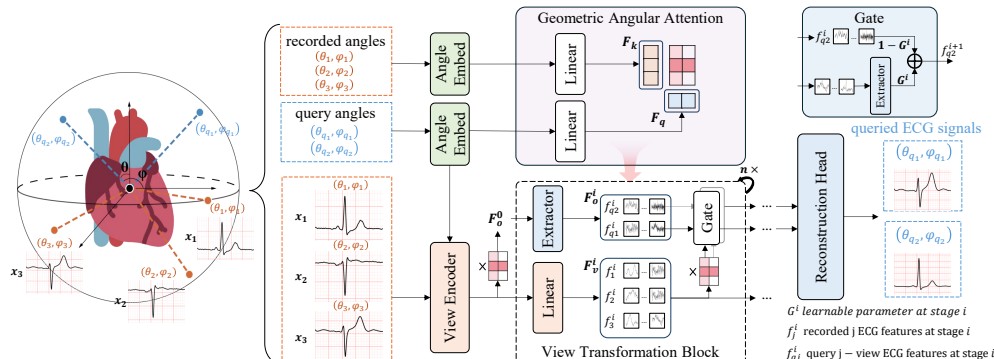

Figure 2: **NEF-NET V2 architecture for Electrocardio Panorama synthesis (illustrated for a 3-input to 2-query view synthesis task as example).** The NEF-NET V2 first employs a View Encoder to extract features from the Recorded ECG that are relevant to the Queried ECG. These extracted features are then fused using a Geometric View Transformer to synthesize the query view.

**Angle Embedding.** We simply extend the "Angular Encoding" module of Nef-Net with an additional linear projection to align feature dimensions, mapping recorded angles $A_k$ and query angles $A_q$ into a higher-dimensional angle space, respectively.

**View Encoder.** Each recorded single-lead ECG signal $x_i \in \mathbb{R}^{1 \times t}$ is processed by a 1-D ResNet basic block following (Chen et al., 2021), yielding $f_{x_i} \in \mathbb{R}^{c \times t'}$. This representation is then concatenated with the query feature $F_q$ in a FiLM-style affine modulation (Perez et al., 2018), which amplifies signal features aligned with $(\theta_q, \varphi_q)$ and suppresses irrelevant ones. Aggregating the outputs from all $l$ recorded leads produces the encoded feature matrix $F_v^0 = [f_1, \ldots, f_l]$.

**Geometric View Transformer (GeoVT).** As each ECG view conveys partially redundant yet complementary information about the query view (Pipberger et al., 1961), our GeoVT is designed with three key components: a Geometric Angular Attention Module ($\text{M}_{\text{GAA}}$), a View Transformation block, and a Reconstruction head. $\text{M}_{\text{GAA}}$ estimates geometric similarity between recorded and query views, the View Transformation block projects recorded features into query-aligned representations, and the Reconstruction head decodes them to reconstruct target-view ECG signals.

(I) *Geometric Angular Attention Module ($\text{M}_{GAA}$).* The $\text{M}_{\text{GAA}}$ implements a cross-attention mechanism (Lin et al., 2022) that compares the angular embedding of the query leads $F_q$ with recorded leads $F_k$. Formally, the Geometric Angular Attention map (GAA) is computed by:

$$GAA = \text{softmax}\left( \frac{F_q W_q (F_k W_k)^\top}{\sqrt{d'}} \right) \tag{1}$$

where $W_q, W_k \in \mathbb{R}^{d \times d'}$ are learnable projection matrices.

(II) *View Transformation Block.* In GeoVT, we stack $L$ view transformation blocks to transfer hierarchical features from recorded signals to the query signals. In block $i$, the recorded signal features $F_v^i$ are projected into an angular latent space by $F_v^{i+1} = \text{Linear}(F_v^i)$ and fused according to the GAA. The resulting representations are integrated block by block through a spatial gating mechanism, by:

$$F_o^{i+1} = F_o^i \odot (1 - G^i) + Ext.(F_v^i \times GAA) \odot G^i \tag{2}$$

where $G^i$ is a learnable parameter with a sigmoid function. The feature extractor (*Ext.*) follows the design of SE blocks (Hu et al., 2018). Through this hierarchical process, GeoVT progressively refines features from the recorded ECG signals in a coarse-to-fine manner, focusing on those relevant to the query view and enabling effective cross-view transformation. Notably, all blocks share the same GAA map as defined in Eq. 1.

(III) *reconstruction head.* The reconstruction head maps the fused embeddings $F_o^L$ back to the time domain using a sequence of upsampling blocks. Each block performs linear interpolation, followed by a convolution module that incorporates a spectral-normalized 1D convolution (Miyato et al., 2018), layer normalization, and a GELU activation.

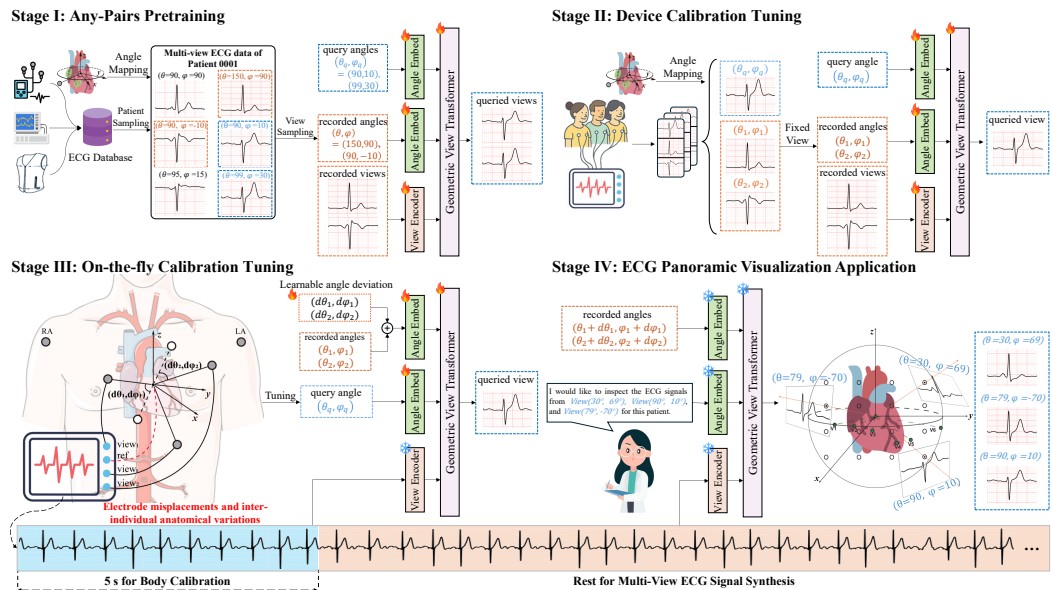

Figure 3: The Development (Stage I, II, III) and Deployment (Stage IV) Workflow of NEF-NET V2.

## 3.2 NEF-NET V2 DEVELOPMENT AND DEPLOYMENT

To deploy an ECG analysis model in real-world settings, a central challenge lies in the heterogeneity of recording devices (Bailey et al., 1990) and the variability arising from operator-dependent procedures and patient-specific variations (Schijvenaars et al., 2008). The original Nef-Net overlooks device-specific discrepancies and case-specific variability. To ensure that our NEF-NET V2 remains robust and clinically applicable, we design a three-stage development and deployment strategy: (1) a device- and case-agnostic pretraining stage, called Any-Pairs Pretraining (ANYPRE), which enables NEF-NET V2 to learn invariant spatiotemporal representations across views under laboratory conditions; (2) Device Calibration (D-CAL), a device-specific calibration stage that adapts NEF-NET V2 to different ECG devices during deployment; and (3) On-the-fly Calibration (OF-CAL), a case-adaptive calibration stage applied at each examination to align the model with case-level variability.

**Stage I: Any-Pairs Pretraining**   In this phase, NEF-NET V2 is pretrained on ECG cases collected from heterogeneous devices to learn fundamental ECG patterns. For each case, the available ECG views are randomly partitioned into two subsets to form the recorded–query pairs $(X_i, Y_i)$. *All* parameters are kept trainable, and we dynamically sample these recorded–query pairs in model training. Notably, in accordance with ECG principles, two limb leads are consistently designated as recorded signals, as they provide essential reference potentials for constructing the remaining leads. Following (Chen et al., 2021), we optimize the network with the mean absolute error (MAE) loss, defined as $\mathcal{L}_{\mathrm{MAE}} = \parallel \hat{Y}_i - Y_i \parallel_1$, where $\hat{Y}_i$ are synthesized by the model.

This strategy exposes the model to diverse sensor characteristics and patient demographics, mitigating protocol-specific overfitting while compelling it to infer cardiac dynamics from arbitrary lead combinations, thereby enhancing robustness and generalization to unseen electrode configurations.

**Stage II: Device Calibration**   Due to variations in ECG acquisition protocols across devices—including hardware design, electrode materials, and others—we present a Device Calibration stage for local adaptation. In this stage, the model is fine-tuned on all recorded–query pairs from the target device using $\mathcal{L}_{\mathrm{MAE}}$, markedly improving alignment with the specific hardware configuration.

**Stage III: On-the-fly Calibration**   Although an ideal viewing angle is defined by (Chen et al., 2021) and this paper, the actual recorded ECG angles $(\theta_{\mathrm{real}}, \varphi_{\mathrm{real}})$ often deviate substantially from the ideal $(\theta, \varphi)$ due to two primary factors: (1) **electrode placement variability** arising across ECG examinations, causing viewing angle deviations; and (2) **inter-subject anatomical variability**, such as differences in heart position, which introduce subject-specific angular offsets from the population mean for each view (*e.g.*, a standard deviation of up to $10.6°$ can be observed in Panobench). ***However, the precise offsets are generally difficult to obtain directly.*** To compensate for these discrepancies,

we introduce learnable angular deviation parameters $(d\theta, d\varphi)$, which are added to the ideal angles to form $(\theta + d\theta, \varphi + d\varphi)$, enabling NEF-NET V2 to dynamically adjust for both sources of variation. In accordance with clinical ECG recording standards (minimum 10-second duration), the initial 5-second segment is allocated for model calibration on the fly. During this stage, the *View Encoder* and *Reconstruction Head* parameters remain frozen, while fine-tuning adapts the angle embeddings to individual-specific deviations.

### 3.3 PANOBENCH: A DENSE BENCHMARK FOR *Electrocardio Panorama*

To enable comprehensive evaluation of panoramic ECG view synthesis, we curated a new benchmark, **Panobench**, which for the first time extends beyond previous datasets limited to only **8** or **12** views. Panobench comprises **4,470** ten-second recordings with **48** viewpoints (**6** limb and 42 precordial leads), each annotated with CT-derived spherical coordinates $(\theta, \varphi)$ following (Chen et al., 2021). By expanding to 48 leads (views), Panobench provides a high-resolution representation of cardiac electrical dynamics, enabling signal analysis from diverse perspectives. The 48-viewpoint signals and their corresponding angular positions are illustrated in Figure 4. This design supports rigorous validation of panoramic ECG synthesis methods and establishes a foundation for clinical translation. In contrast, traditional **12**-lead settings require input, supervision, and synthesis to be partitioned among the same limited leads, making validation inherently less comprehensive.

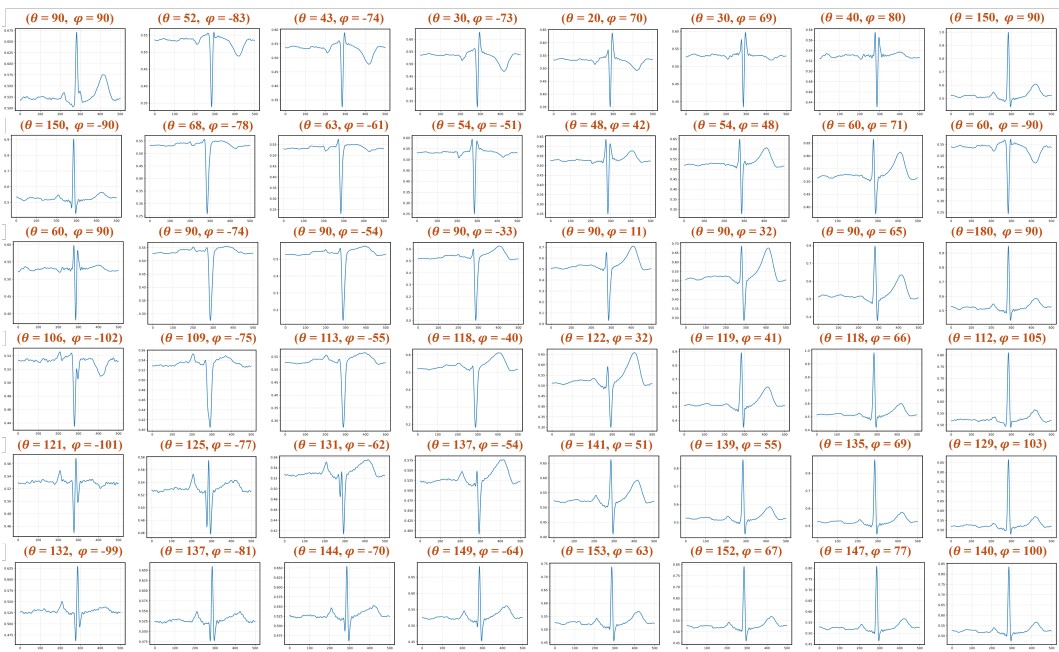

Figure 4: A representative example from Panobench, illustrating the 48 distinct ECG view. Each view corresponds to a unique angular position.

## 4 EXPERIMENTS

### 4.1 DATASETS AND IMPLEMENTATION DETAILS

We conduct experiments on the PTB-XL dataset (Wagner et al., 2020), Tianchi ECG dataset[3], CPSC2018 dataset (Liu et al., 2018), ChinaDB dataset (Zheng et al., 2020), and our curated dataset-Panobench. All ECG recordings were resampled to 500 Hz. Each dataset is randomly split into 80 percent training and 20 percent testing. Detailed information about the datasets is provided in Appendix C.

In the *Any-Pairs Pretraining* stage, the model is trained on the combined training sets of four public ECG datasets, with evaluation conducted on their respective test sets. In the *Device Calibration* stage,

---

[3]https://tianchi.aliyun.com/competition/entrance/231754/information?lang=en-us

Table 2: Performance on *reconstruction* and *synthesis* tasks on the Tianchi, ChinaDB, CPSC2018 and PTB-XL datasets. In the synthesis tasks, the numbers of views for input, reconstruction, and synthesis are orderly listed in parentheses. The best performances are highlighted in **bold**.

| | | ChinaDB | | CPSC2018 | | Tianchi | | PTB-XL | |
|---|---|---|---|---|---|---|---|---|---|
| | | PSNR | SSIM | PSNR | SSIM | PSNR | SSIM | PSNR | SSIM |
| *View Reconstruction* | | | | | | | | | |
| E-LSTM | (3,9) | 20.56±0.38 | 0.811 | 21.37±0.41 | 0.824 | 22.76±0.43 | 0.848 | 20.04±0.43 | 0.810 |
| ECGRecover | (3,9) | 30.47±0.26 | 0.958 | 30.12±0.31 | 0.966 | 31.47±0.37 | 0.971 | 28.57±0.31 | 0.942 |
| EKGAN | (3,9) | 32.76±0.31 | 0.967 | 33.35±0.24 | 0.975 | 34.39±0.29 | 0.977 | 31.71±0.28 | 0.972 |
| SSSD | (3,9) | 32.53±0.37 | 0.966 | 32.67±0.39 | 0.972 | 33.59±0.41 | 0.975 | 31.42±0.47 | 0.972 |
| Nef-Net | (3,9) | 29.59±0.20 | 0.961 | 29.12± 0.26 | 0.958 | 31.44±0.21 | 0.965 | 30.22± 0.27 | 0.962 |
| NEF-NET V2 | (3,9) | **35.84**±0.22 | **0.977** | **36.12**±0.38 | **0.981** | **37.13**±0.44 | **0.982** | **35.21**±0.36 | **0.974** |
| KIM | (8,12) | 27.65 | 0.952 | 27.82 | 0.956 | 28.01 | 0.937 | 26.71 | 0.929 |
| ECGRecover | (8,12) | 38.96±0.59 | **0.983** | 37.74±0.73 | **0.985** | 39.72±0.41 | **0.986** | 35.92±0.79 | **0.985** |
| EKGAN | (8,12) | 39.43±0.52 | 0.976 | **39.56**±0.49 | 0.976 | 40.37±0.71 | 0.981 | **39.83**±0.82 | 0.983 |
| SSSD | (8,12) | 39.37±0.35 | 0.978 | 38.21±0.27 | 0.980 | 40.57±0.38 | 0.983 | 38.37±0.47 | 0.980 |
| Nef-Net | (8,12) | 32.74± 0.31 | 0.967 | 31.68± 0.33 | 0.971 | 33.72± 0.29 | 0.961 | 30.58±0.30 | 0.977 |
| NEF-NET V2 | (8,12) | **39.54**±0.37 | 0.978 | 38.69±0.45 | 0.981 | **41.52**±0.62 | 0.983 | 39.15±0.41 | 0.983 |
| *Unseen View Synthesis* | | | | | | | | | |
| Nef-Net | (3,8,1) | 25.24±0.29 | 0.951 | 26.72±0.25 | 0.957 | 27.92±0.27 | 0.959 | 24.10±0.22 | 0.922 |
| NEF-NET V2 | (3,8,1) | **32.57**±0.21 | **0.981** | **33.62**±0.17 | **0.985** | **34.46**±0.35 | **0.976** | **33.41**±0.29 | **0.982** |
| Nef-Net | (5,6,1) | 26.06±0.31 | 0.954 | 26.11±0.33 | 0.948 | 28.01±0.30 | 0.959 | 25.37±0.34 | 0.942 |
| NEF-NET V2 | (5,6,1) | **33.16**±0.30 | **0.982** | **32.76**±0.27 | **0.982** | **34.82**±0.41 | **0.977** | **32.07**±0.35 | **0.986** |

a single dataset with fixed input-lead configurations is utilized for both training and testing. In the *On-the-fly Calibration* stage, the first 5-second segment of each patient's ECG recording is used for model adaptation, with the subsequent 5-second segment reserved for performance evaluation.

All experiments are implemented using PyTorch 1.9 on three NVIDIA RTX2080Ti GPUs, each with 11 GB of memory. The implementation details of each experiment are shown in Appendix D.

## 4.2 PERFORMANCE EVALUATION

In this section, we evaluate NEF-NET V2's ability to generate an *Electrocardio Panorama* via two complementary tasks: *reconstruction*, in which the model regenerates ECG signals from viewpoints used in training, and *synthesis*, in which it produces signals for entirely unseen viewpoints, following (Chen et al., 2021). Signal quality is assessed using the peak signal-to-noise ratio (PSNR) and the structural similarity index (SSIM). These established metrics were chosen to ensure a fair and direct comparison with the Nef-Net baseline. All experiments were evaluated on arbitrary-length ECG segments. For the *reconstruction* task, Nef-Net (Chen et al., 2021), E-LSTM (Sohn et al., 2020), and KIM (Kors et al., 1990) were evaluated using the lead configuration settings adopted from previous work (Chen et al., 2021). ECGRecover (Lence et al., 2025), EKGAN (Joo et al., 2023), and SSSD (Alcaraz & Strodthoff, 2022) were evaluated under the setting of 3-to-9 and 8-to-12 reconstruction settings. *Owing to the use of longer ECG recordings rather than beat-level data, our Nef-Net reproduction shows degraded performance compared to the original.* For the *synthesis* task, we compare only with Nef-Net, as it is the only existing method capable of generating signals for unseen viewpoints. The number of views for input, supervision (reconstruction), and synthesis are listed in parentheses, in that order. All experiments were repeated multiple times for validation. We will report the standard deviation for PSNR data, while the standard deviation for SSIM is omitted due to its negligible variation.

As shown in Table 2, our NEF-NET V2 significantly outperforms previous methods on the *reconstruction* task under the 3-to-9-lead setting. These results provide a strong reference for its *synthesis* capability. While under the more challenging 8-to-12-lead setting, the performance of existing state-of-the-art models is closely matched.

The lower portion of Table 2 reveals a performance gap between the *synthesis* and *reconstruction* capabilities of both NEF-NET V2 and Nef-Net, highlighting a fundamental challenge in generalizing to novel ECG views. Despite this common challenge, NEF-NET V2 consistently surpasses Nef-Net across all datasets, achieving substantially higher PSNR and SSIM in *Electrocardio Panorama* synthesis. Our investigation on the Panobench dataset identified a key factor underlying these performance gaps, with a detailed analysis provided in Section 4.3.

Table 3: Performance Comparison of Nef-Net and NEF-NET V2 on ECG Panoramic Synthesis with Varying Input and Supervised Leads on Panobench. **Lead Types:** Bipolar limb leads (I, II) measure potential between two limb electrodes. Unipolar leads (view-18, view-23) measure potential between body electrode and heart reference. Better values in **bold**.

| Input Leads | Method | Supervised Leads | | | | | | | |
|---|---|---|---|---|---|---|---|---|---|
| | | 3 | | 6 | | 9 | | 12 | |
| | | PSNR | SSIM | PSNR | SSIM | PSNR | SSIM | PSNR | SSIM |
| I, II, view-18 | Nef-Net (Rec) | 35.13±0.35 | 0.983 | 34.22±0.41 | 0.981 | 34.28± 0.49 | 0.983 | 34.68±0.42 | 0.981 |
| | NEF-NET V2 (Rec) | **37.25**±0.40 | **0.989** | **36.22**± 0.20 | **0.985** | **36.15**±0.22 | **0.985** | **36.06**±0.68 | **0.985** |
| | Nef-Net (Syn) | 21.13±1.15 | 0.894 | 29.49±0.37 | 0.965 | 31.89± 0.33 | 0.973 | 32.98±0.39 | 0.978 |
| | NEF-NET V2 (Syn) | **31.63**±0.43 | **0.973** | **33.05**± 0.31 | **0.973** | **35.13**±0.47 | **0.983** | **35.57**±0.23 | **0.983** |
| I, II, view-18, 24, 31 | Nef-Net (Rec) | 35.77±0.25 | 0.986 | 34.80±0.43 | 0.974 | 34.19± 0.57 | 0.976 | 35.14±0.61 | 0.978 |
| | NEF-NET V2 (Rec) | **38.52**±0.31 | **0.992** | **38.18**± 0.34 | **0.984** | **36.66**±0.28 | **0.987** | **38.56**±0.57 | **0.987** |
| | Nef-Net (Syn) | 21.08±0.70 | 0.880 | 30.35±0.41 | 0.965 | 32.15± 0.36 | 0.973 | 33.83± 0.27 | 0.977 |
| | NEF-NET V2 (Syn) | **32.01**±0.37 | **0.972** | **33.24**± 0.28 | **0.972** | **35.06**± 0.25 | **0.980** | **36.11**±0.40 | **0.982** |
| I, II, view-18, 24, 31, 37, 40 | Nef-Net (Rec) | 35.82±0.56 | 0.984 | 32.37±0.41 | 0.975 | 33.86± 0.68 | 0.982 | 35.34±0.84 | 0.982 |
| | NEF-NET V2 (Rec) | **40.85**±0.55 | **0.995** | **37.25**± 0.42 | **0.988** | **36.68**±0.34 | **0.987** | **38.48**±0.68 | **0.988** |
| | Nef-Net (Syn) | 21.78±1.04 | 0.917 | 28.06±0.39 | 0.964 | 31.94± 0.37 | 0.976 | 34.47±0.25 | 0.980 |
| | NEF-NET V2 (Syn) | **32.86**± 0.39 | **0.973** | **33.21**± 0.24 | **0.975** | **34.57**±0.13 | **0.979** | **36.11**± 0.35 | **0.985** |

## 4.3 ELECTROCARDIO PANORAMA SYNTHESIS EVALUATION ON PANOBENCH

Most existing datasets provide only 12 views, which constrains comprehensive evaluation of panoramic synthesis. Therefore, we curated Panobench to more thoroughly assess the effectiveness of NEF-NET V2 in synthesizing Electrocardio Panorama. Our evaluation is designed to answer three questions: 1) How does NEF-NET V2 compare with Nef-Net in panoramic ECG synthesis? 2) How does the number of input leads affect reconstruction and synthesis performance? 3) How does the number of supervised leads affect reconstruction and synthesis performance?

Given the extensive combinations of input and target views, we align our input configuration with the panorama training settings used for standard 12-lead ECG synthesis. Because the combinations are numerous, we report representative cases in Table 3. Specifically, we select limb lead I and limb lead II as inputs, together with precordial-like signals corresponding to the chest-lead view angles.

Figure 5 compares the synthesis and reconstruction performance of NEF-NET V2 and Nef-Net under varying levels of supervision. Notably, NEF-NET V2 achieves higher accuracy in both tasks. Across both models, reconstruction performance serves as an upper bound, quantifying the gap between each model's synthesis capability and its inherent representational limits. We observe that synthesis performance improves markedly with more supervised leads in both models, while reconstruction remains stable. This narrowing performance gap indicates enhanced generalization, as the model's synthesis capability approaches its representational upper bound. In contrast, Nef-Net struggles under sparse-lead supervision but gradually improves as more leads are available, eventually approaching NEF-NET V2 's synthesis performance. In contrast to the significant role of supervision, the variation in the number of input leads (from 3 to 7) did not yield observable changes in the reconstruction or synthesis performance of either model.

The inferior performance of Nef-Net under sparse supervision stems from its limited use of view-specific information. Its encoder-decoder design compresses signals into a latent cardiac field and uniformly averages recorded features, neglecting the varying physiological relevance of individual views. As a result, query-irrelevant signals are mixed into the representation, leading to blurred morphology and degraded fidelity, which becomes more pronounced with fewer supervised leads. In contrast, NEF-NET V2 reformulates ECG synthesis as a direct view-

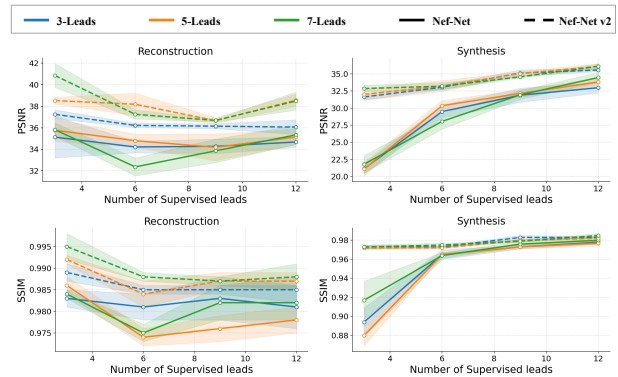

Figure 5: Reconstruction and Synthesis Under Varying Inputs and Supervision.

to-view transformation. By leveraging angular relationships, it explicitly models inter-lead dependencies and selectively amplifies geometry-consistent features. This design preserves diagnostically relevant details and sustains high synthesis accuracy, even under sparse-lead supervision.

## 4.4 PERFORMANCE ACROSS DISEASE CATEGORIES

Multi-lead ECGs capture cardiac electrical activity from spatially distinct perspectives (*e.g.*, chest leads V1–V4 reflect the anterior wall). To rigorously assess whether NEF-NET V2 can synthesize diagnostically reliable signals across diverse cardiac conditions, we evaluate it on the CPSC2018 dataset, which encompasses nine categories of cardiac disorders, with detailed sample distributions provided in Appendix C.

Quantitative results in Table 4 demonstrate NEF-NET V2's superior reconstruction fidelity across all pathological categories, with an average PSNR improvement of 6.9 dB and consistent SSIM gains. Notably, its performance on Atrial Fibrillation (AF) signals improved by 7.3 dB. Beyond raw fidelity, the consistently larger margins on pathological cases (*e.g.*, AF, I-AVB, PVC, STE) indicate that NEF-NET V2 may not only synthesize general signal morphology but also preserve key pathological signatures. The reduced performance gap between normal and abnormal rhythms suggests enhanced disease-adaptive capability, underscoring the potential of NEF-NET V2 to generate clinically reliable ECG waveforms under diverse diagnostic scenarios.

Table 4: View *Synthesis* Performance on CPSC2018 across different diseases. Better in **bold**.

|  |  | Normal | AF | I-AVB | LBBB | RBBB | PAC | PVC | STD | STE | AV |
|---|---|---|---|---|---|---|---|---|---|---|---|
| Nef-Net | PSNR | 30.51 | 25.12 | 27.35 | 25.24 | 25.19 | 27.64 | 28.10 | 28.95 | 27.15 | 26.72 |
| NEF-NET V2 | PSNR | **35.41** | **32.42** | **33.41** | **28.35** | **32.18** | **33.67** | **33.14** | **34.96** | **33.24** | **33.62** |
| Nef-Net | SSIM | 0.977 | 0.941 | 0.961 | 0.944 | 0.943 | 0.959 | 0.962 | 0.965 | 0.958 | 0.957 |
| NEF-NET V2 | SSIM | **0.989** | **0.976** | **0.978** | **0.955** | **0.975** | **0.976** | **0.982** | **0.984** | **0.977** | **0.985** |

The clinical value of NEF-NET V2 lies in its ability to observe cardiac electrophysiology from arbitrary viewpoints, making the preservation of pathological features in the synthesized signals critically important. To verify this, we visualize synthesized signals at pathology-relevant viewpoints using real LBBB and RBBB cases. As these pathological characteristics typically manifest most prominently in the V1 and V6 leads, they provide the most appropriate basis for assessing the clinical relevance of NEF-NET V2's outputs. Due to space constraints, we present this analysis in Appendix E.

## 4.5 ABLATION STUDY FOR THE THREE-STAGE DEVELOPMENT FRAMEWORK

The Any-Pairs Pretraining stage establishes a robust baseline by learning from large-scale ECG datasets. The Device Calibration stage adapts the model to specific ECG acquisition devices, while the On-the-fly Calibration stage provides case-level refinements by correcting for electrode placement variability and subject-specific anatomy. As shown in Table 5, incorporating a Device Calibration step yields consistent PSNR gains of 0-1.07 dB and SSIM improvements of 0-0.004 across benchmarks, while adding On-the-fly Calibration Calibration achieves comparable enhancements at 1.75-2.74 on PSNR and 0.001-0.010 on SSIM. This indicates that inter-subject anatomical differences have a more pronounced impact on Electrocardio Panorama quality than inter-device variations. To further investigate this, we analyze the impact of electrode placement offsets on NEF-NET V2's synthesis capability, with detailed results provided in Appendix E.

Table 5: The impact of different stage for view synthesis tasks on the Tianchi, ChinaDB, CPSC2018 and PTB-XL datasets.

|  | ChinaDB | | CPSC2018 | | Tianchi | | PTB-XL | |
|---|---|---|---|---|---|---|---|---|
|  | PSNR | SSIM | PSNR | SSIM | PSNR | SSIM | PSNR | SSIM |
| Any-Pairs Pretraining | 29.83±0.20 | 0.972 | 31.01±0.23 | 0.975 | 32.71±0.21 | 0.975 | 31.15±0.23 | 0.981 |
| Device Calibration | 30.77±0.14 | 0.973 | 32.08±0.17 | 0.979 | 33.05±0.12 | 0.977 | 31.15 ±0.11 | 0.981 |
| On-the-fly Calibration | 32.57±0.21 | 0.981 | 33.62± 0.17 | 0.985 | 34.46±0.35 | 0.976 | 33.41±0.29 | 0.982 |

To further illustrate the benefits of each stage, Figure 6 shows the progressive enhancements achieved by our three-stage framework. Importantly, subtle variations in ECG signals are clinically critical, as

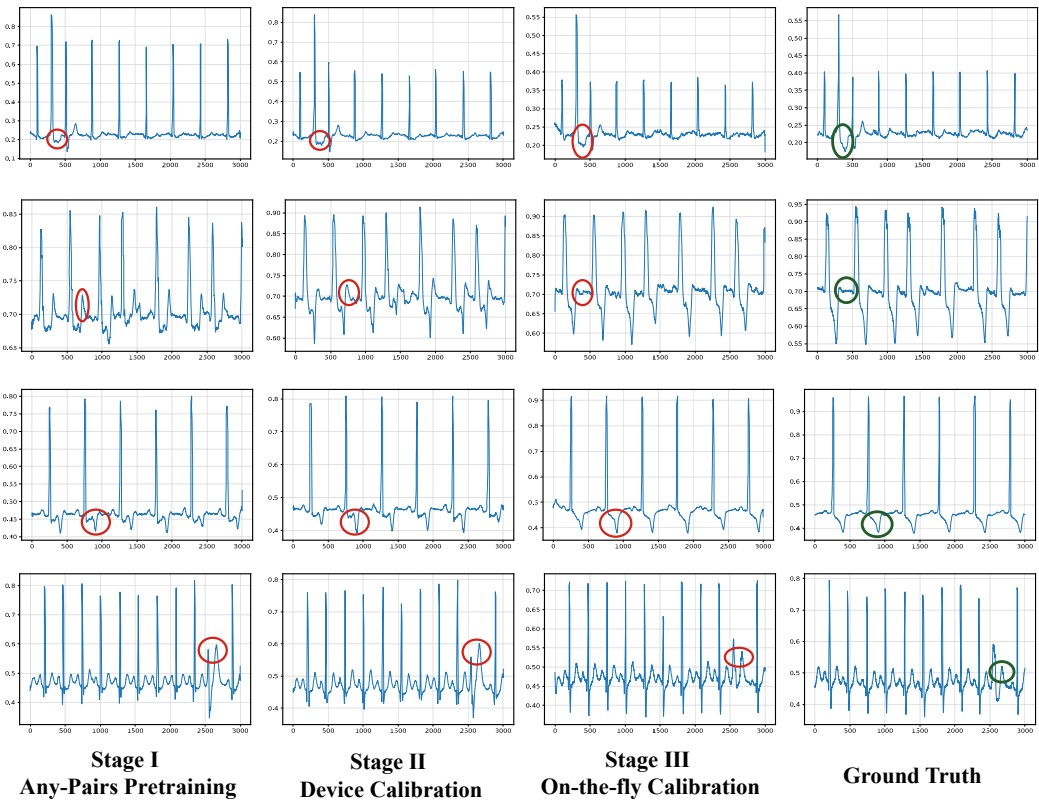

Figure 6: Representative examples from CPSC2018 illustrate progressive synthesis of the V5 view by NEF-NET V2 across training stages. Green circles mark clinically relevant diagnostic details in the real signals, whereas red circles highlight their increasingly accurate recovery as our multi-stage training progresses.

even minor waveform differences may correspond to arrhythmia events or pathognomonic ST-segment changes. Consequently, even modest improvements in PSNR directly reflect the preservation of clinically salient waveform characteristics that are indispensable for accurate diagnosis and treatment. For instance, in Fig. 6 (fourth row), the heartbeat between samples 2300–2800 shows that the actual signal has a T-wave amplitude smaller than the R-wave, while the synthesized signals at Stage I and Stage II exhibit an ST segment higher than the R wave, a morphology that in clinical practice may indicate acute myocardial infarction, thereby potentially misleading diagnosis. The final output after On-the-fly Calibration accurately reproduces the correct relationship, resulting in a diagnosable-level synthesis. These findings demonstrate the necessity of the complete development-to-deployment framework: Any-Pairs Pretraining learns generalizable cardiac spatial priors, Device Calibration addresses device-level heterogeneity, and On-the-fly Calibration adapts to subject-specific anatomical variation. Together, these stages close the gap between controlled training conditions and the demands of real-world clinical deployment.

## 5 CONCLUSIONS

This work advances *Electrocardio Panorama* synthesis from controlled experimental settings to real-world applications. Our key methodological contribution is to reformulate ECG view synthesis, motivated by heart vector theory and the limitations of Nef-Net's feature averaging as a direct view-to-view transformation problem. Building on this formulation, we introduce a three-stage development pipeline: large-scale pretraining, device-specific calibration, and On-the-fly Calibration to case-specific electrode-induced viewpoint shifts. To enable rigorous evaluation, we curate Panobench, the first 48-lead ECG dataset annotated with precise CT-derived spherical coordinates $(\theta, \varphi)$, establishing a comprehensive benchmark for *Electrocardio Panorama* synthesis. Experiments demonstrate that our NEF-NET V2 consistently outperforms previous works by a substantial margin.

ACKNOWLEDGMENTS

This work was partly supported by the Guangdong Basic and Applied Basic Research Foundation (2026A1515011793), the Youth S&T Talent Support Programme of Guangdong Provincial Association for Science and Technology (SKXRC2025467), as well as the National Key R&D Program of China (2023YFE0110200).

ETHICAL STATEMENT

The collection of the Panobench dataset complied with standard ethical protocols, and written informed consent was obtained from all participants, granting explicit permission for the academic use. All data were rigorously anonymized to safeguard participant privacy.

REPRODUCIBILITY STATEMENT

To ensure reproducibility, we provide full experimental details in the Appendix.

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

## A LLM USAGE

We made light use of a large language model (GPT-4, OpenAI) for grammar checking, spelling correction, and minor language polishing. The model was **NOT** involved in research ideation, experimental design, analysis, or interpretation. All scientific contributions and conclusions are entirely the responsibility of the authors.

## B THEORETICAL FOUNDATIONS FOR MULTI-VIEW ECG GENERATION

Electrocardiographic (ECG) signals represent the temporal dynamics of cardiac depolarization and repolarization, recorded as time-series waveforms. A typical cardiac cycle consists of six characteristic deflections: the P wave, PR segment, QRS complex, ST segment, T wave, and TP segment. These deflections reflect distinct physiological processes. For example, the P wave corresponds to atrial depolarization, the QRS complex captures rapid ventricular depolarization, and the T wave reflects ventricular repolarization. Clinically, subtle changes in the amplitude, duration, or morphology of these components can serve as critical biomarkers, such as ST-segment elevation indicating acute myocardial infarction or QRS widening suggesting conduction abnormalities (*e.g.*, bundle branch block). This fundamental structure underlies all lead systems and provides the physiological basis for both diagnostic interpretation and computational modeling.

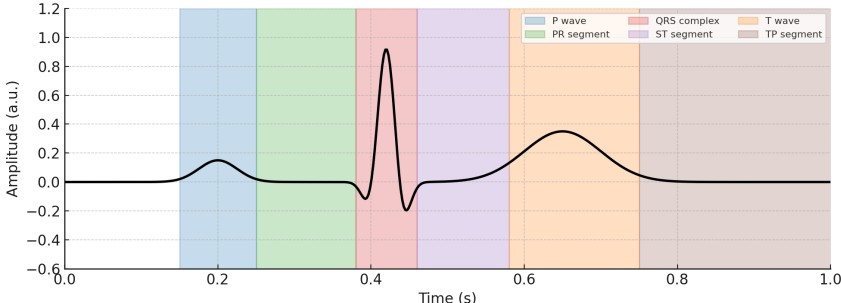

Figure 7: Standard ECG waveform with six characteristic components highlighted: P wave, PR segment, QRS complex, ST segment, T wave, and TP segment.

In clinical cardiology, ECG signals are broadly categorized into two lead systems: the *bipolar* and *unipolar* lead systems. The bipolar system—comprising Leads I, II, III, and the augmented limb leads $aVR$, $aVL$, and $aVF$—records the potential differences between pairs of electrodes placed on the limbs. These leads provide a global yet relatively coarse projection of the heart's electrical activity onto the frontal plane, offering complementary directional information about the cardiac depolarization vector.

In contrast, the unipolar lead system, such as chest leads $V1, V2, V3, \ldots$ and experimental leads like `view-1`, `view-2` in Panobench, record the electrical potential at a specific anatomical location relative to a reference point (often Wilson's central terminal). The Wilson central terminal, defined as the average of the three limb electrodes, approximates the cardiac reference potential:

$$V_{\text{WCT}} = \frac{V_{\text{RA}} + V_{\text{LA}} + V_{\text{LL}}}{3} \tag{3}$$

This configuration is conceptually analogous to a camera network in computer vision, in which each camera captures a distinct projection of a three-dimensional scene. Similarly, each ECG lead acts as a biological "camera", recording a unique spatial projection of the heart's dynamic electrical activity over time. In standard clinical practice, for example, the precordial leads V1–V4 primarily assess the anterior wall of the heart, whereas V5–V6 provide information about the lateral wall (Anderson et al., 1994; Case et al., 1979). However, the fixed geometry of the conventional 12-lead ECG does not always capture all diagnostically relevant patterns. Clinicians frequently supplement the standard configuration with additional non-standard leads based on personal interpretation styles or case-specific requirements (Zhan et al., 2024). These observations underscore the inherent importance of examining cardiac electrical activity from multiple viewpoints.

**Theoretical Foundations of Neural Electrical Field for ECG Synthesis.** From an electrophysiological perspective, the cardiac electrical activity can be modeled as a time-varying source whose field is described by a multipolar expansion. As shown in Eq. 4, the extracellular potential $V(x, y, z)$ can be expressed as a series of dipole, quadrupole, and higher-order terms. In clinical practice, higher-order contributions are negligible at the body surface, leading to a far-field dipole approximation.

$$V(x, y, z) = \frac{1}{4\pi\sigma} \left( \frac{\sum p_i \cdot \hat{r}}{r^2} + \frac{\sum [(p_i \cdot r_i')\hat{r} - p_i(r_i' \cdot \hat{r})]}{r^3} + \cdots \right)$$
$$\approx \frac{1}{4\pi\sigma} \frac{\sum p_i \cdot \hat{r}}{r^2} \tag{4}$$
$$\approx \frac{1}{4\pi\sigma} \frac{p_x(x - x_0) + p_y(y - y_0) + p_z(z - z_0)}{[(x - x_0)^2 + (y - y_0)^2 + (z - z_0)^2]^{3/2}},$$

where $\sigma$ denotes tissue conductivity, $(p_x, p_y, p_z)$ are the dipole strengths along the cartesian axes, and $(x_0, y_0, z_0)$ denotes the geometric center of cardiac activity, which is not directly measurable. In clinical ECG acquisition, the Wilson central terminal is commonly adopted as a practical approximation to this reference potential. This formulation provides the theoretical underpinning for ECG view synthesis: from a limited set of recorded leads, one can infer the cardiac dipole and reconstruct signals for any lead position $(x, y, z)$ or, equivalently, angular coordinates $(\theta, \varphi)$.

Building on this, Nef-Net uses the relative angles between input and query leads as spatial priors to implicitly model the cardiac electrical field and synthesize the corresponding signals. Within this framework, the network parameters learn implicit representations of thoracic conductivity and field properties, enabling a physiologically grounded reconstruction of signals from unobserved viewpoints.

**Theoretical Foundations of View Transformation for ECG Synthesis.** From the perspective of cardiac vector theory (Grant, 1950), the cardiac electrophysiological field can be represented as a three-dimensional spatial vector. ECG signals from different leads can be viewed as projections of this vector from various viewpoints. Therefore, a small subset of independent leads suffices to estimate the global electric field, such that the potential measured at lead $i$ can be expressed as:

$$V_i(t) \approx \mathbf{p}(t) \cdot \hat{r}_i, \tag{5}$$

where $\mathbf{p}(t)$ is the time-varying cardiac dipole vector and $\hat{r}_i$ is the orientation of lead $i$. This formulation highlights that multi-lead ECGs represent observations of the cardiac electrical field from different viewpoints, with varying degrees of information redundancy between these views. Synthesizing a novel lead $j$ from recorded leads $\{i\}$ can therefore be viewed as learning a transformation conditioned on angular relationships between $\hat{r}_i$ and $\hat{r}_j$:

$$V_j(t) \approx \mathcal{T}\big(V_i(t), \hat{r}_i, \hat{r}_j\big), \tag{6}$$

Let $\mathcal{T}$ denote the projection operator implied by the dipole model. This formulation suggests that ECG signals from recorded views can be transformed to a target view through $\mathcal{T}$, bypassing the need for explicit reconstruction of the underlying cardiac electrical field.

Our proposed NEF-NET V2 implements this insight by mapping angular coordinates $(\theta, \varphi)$ into a learnable representation, which enables feature projection based on spatial-angular relationships between recorded leads and target leads. By doing so, the network learns to approximate the theoretical operator $\mathcal{T}$, effectively performing direct view-to-view transformation.

## C  DATASET DESCRIPTION

We evaluate NEF-NET V2 on several widely used public ECG benchmarks. **PTB-XL** contains 21,837 ten-second 12-lead ECGs sampled at 500 Hz and covering a wide range of cardiac pathologies. **Tianchi** consists of 31,779 12-lead recordings, also sampled at 500 Hz. **ChinaDB** provides 10,646 ten-second ECGs (5,000 samples each) at 500 Hz. **CPSC2018** includes 6,877 multi-hospital ECGs with durations ranging from 6–60 seconds, sampled at 500 Hz. We use CPSC2018 to assess the diagnostic reliability of synthesized signals across diverse pathological conditions; the distribution across nine diagnostic classes is summarized in Table 7.

The *Panobench* dataset is a self-collected panoramic ECG resource comprising 4,470 ten-second recordings sampled at 250 Hz. Each sample contains 48 leads (6 limb and 42 precordial) collected under resting conditions from subjects aged 18–28. Electrode positions were manually annotated on CT volumes to obtain precise spherical coordinates for all leads. Compared with existing public datasets (*e.g.*, PTB-XL and CPSC2018), Panobench offers a substantially denser set of ECG viewpoints with explicit angular annotations, providing a unique foundation for evaluating panoramic ECG synthesis.

A comprehensive comparison of dataset characteristics is provided in Table 6.

Table 6: Detailed Description of the Used ECG Datasets.

| Dataset | Recordings | Sampling Rate | Duration | Lead |
|---|---|---|---|---|
| PTB-XL | 21837 | 500 Hz/250 Hz | 10s | 12 |
| Tianchi | 31779 | 500 Hz | 10s | 12 |
| CPSC2018 | 6877 | 500 Hz | 6-60s | 12 |
| ChinaDB | 10646 | 500 Hz | 10s | 12 |
| Panobench | 4470 | 250 Hz | 10s | 48 |

Table 7: The data description of CPSC2018

| Class | Training set | % | Testing set | % | Total | % |
|---|---|---|---|---|---|---|
| Normal | 734 | 10.6% | 184 | 2.7% | 918 | 13.3% |
| AF | 878 | 12.8% | 220 | 3.2% | 1098 | 16.0% |
| I-AVB | 563 | 8.2% | 141 | 2.1% | 704 | 10.3% |
| LBBB | 165 | 2.4% | 42 | 0.6% | 207 | 3.0% |
| RBBB | 1356 | 19.7% | 339 | 4.9% | 1695 | 24.6% |
| PAC | 445 | 6.5% | 111 | 1.6% | 556 | 8.1% |
| PVC | 538 | 7.8% | 134 | 1.9% | 672 | 9.7% |
| STD | 660 | 9.6% | 165 | 2.4% | 825 | 12.0% |
| STE | 162 | 2.4% | 40 | 0.6% | 202 | 3.0% |
| Total | 5501 | 80.0% | 1376 | 20.0% | 6877 | 100.0% |

In this work, lead positions were manually annotated using the ITK-SNAP software[4] (Yushkevich et al., 2006). The annotation results are visualized in Figure 8: the three-dimensional rendering (lower-left panel) highlights the segmented cardiac volume (red, used as the reference center), the annotated Panobench leads (blue), and the standard 12-lead precordial electrodes (green).

A subset of 30 subjects was used for detailed analysis of lead orientations. Table 8 reports the mean angular positions derived from these annotations. The averaged limb-lead orientations are: Lead I $(90°, 90°)$, Lead II $(150°, 90°)$, Lead III $(150°, -90°)$, aVR $(60°, -90°)$, aVL $(60°, 90°)$, and aVF $(180°, 90°)$.

Table 8: Angles for Panobench

| RA | $\theta, \varphi$ | | | | | | | | | | | | | | | | LA | $\theta, \varphi$ |
|---|---|---|---|---|---|---|---|---|---|---|---|---|---|---|---|---|---|---|
| | | 4 | 52,-83 | 10 | 43,-74 | 16 | 30,-73 | | | 22 | 20,70 | 28 | 30,69 | 34 | 40,80 | | |
| | | 5 | 68,-78 | 11 | 63,-61 | 17 | 54,-51 | | | 23 | 48,42 | 29 | 54,48 | 35 | 60,71 | | |
| | | 6 | 90,-74 | 12 | 90,-54 | 18 | 90,-33 | Median | | 24 | 90,11 | 30 | 90,32 | 36 | 90,65 | | |
| 1 | 106,-102 | 7 | 109,-75 | 13 | 113,-55 | 19 | 118,-40 | | | 25 | 122,32 | 31 | 119,41 | 37 | 117,66 | 40 | 112,105 |
| 2 | 121,-101 | 8 | 125,-77 | 14 | 131,-62 | 20 | 137,-54 | | | 26 | 141,51 | 32 | 139,55 | 38 | 135,69 | 41 | 129,103 |
| 3 | 132,-99 | 9 | 137,-81 | 15 | 144,-70 | 21 | 149,-64 | | | 27 | 153,63 | 33 | 152,67 | 39 | 147,77 | 42 | 140,100 |

---

[4]http://www.itksnap.org

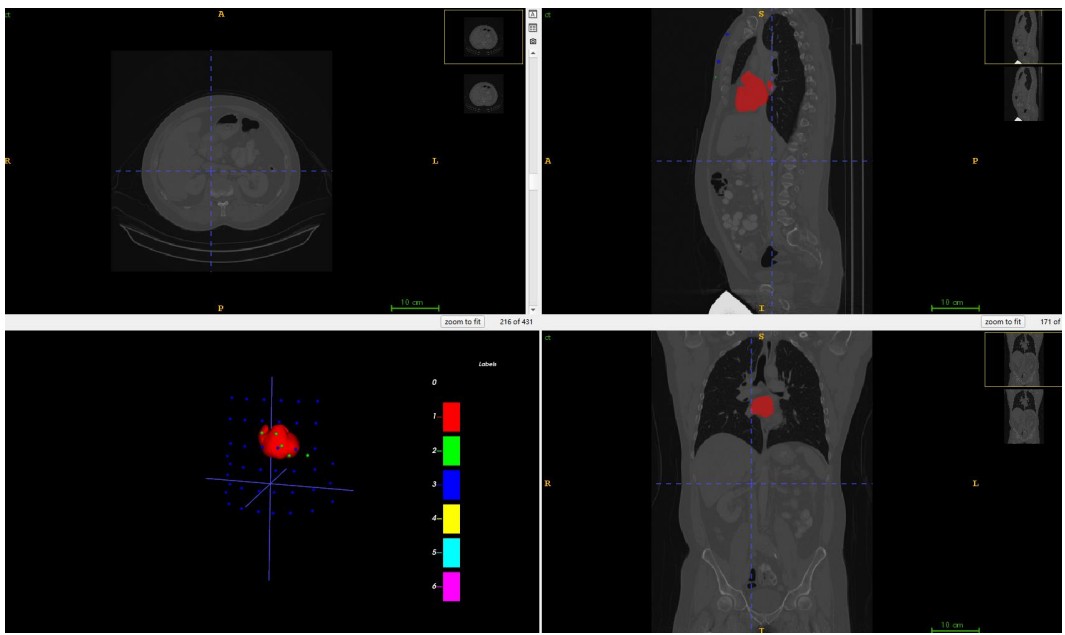

Figure 8: CT Annotation of 48-Lead ECG Electrodes, Six Precordial Leads, and Cardiac Position

## D   IMPLEMENTATION DETAILS OF DIFFERENT EXPERIMENTS

For easy re-implementation, this section documents the experimental configurations and implementation details.

- *Number of view transformer layers: 4;*
- *0ptimizer: AdamW;*
- *Batch size: 32;*
- *Weight decay: $10^{-2}$;*
- *Learning rate step: [50,100,150];*
- *Gamma (a MultiStepLR hype-parameter): 0.5;*

### D.1   ANY-PAIRS PRETRAINING

200-epoch self-supervised phase employing randomly selected lead combinations (3, 4, or 5 leads per input, excluding synthesized chest leads).

- *Training parameters: During training, all model parameters are optimized except for $\alpha$, which is kept frozen;*
- *Learning rate: $10^{-3}$;*
- *Training Datasets: All datasets;*
- *Training epochs: 200*

### D.2   DEVICE CALIBRATION

- *Training parameters: During training, all model parameters are optimized except for $\alpha$, which is kept frozen;*
- *Learning rate: $5 * 10^{-4}$;*
- *Training Datasets: Specific dataset;*
- *Training epochs: 200*

### D.3 ON-THE-FLY CALIBRATION

- *Training parameters: During training, the View Embed block and Reconstruction Head are kept frozen, and only the remaining angle-related parameters are optimized;*
- *Learning rate:* $5 * 10^{-5}$;
- *Training Datasets: Per-person;*
- *Finetune iterations: 100*

## E FUTHER DISCUSSION OF THIS WORK

### E.1 IMPACT OF ON-THE-FLY CALIBRATION ON DATA EFFICIENCY FOR VIEW SYNTHESIS

A data-driven approach is employed to develop the proposed view transformation algorithm, wherein an Any-Pairs pretraining strategy is introduced to enable the model to internalize the "language of ECG signals" while effectively leveraging heterogeneous ECG datasets with diverse lead configurations. Such pretraining is critical for scaling data-driven models to robustly capture cross-lead correlations and improve generalization across varied acquisition settings.

To quantify its impact, we conducted a data-efficiency study by varying the proportion of training data on the CPSC2018 dataset, both with and without pretraining (Table 9). Results demonstrate that in low-data regimes (1 percent and 5 percent), models trained from scratch exhibit severe performance degradation (e.g., PSNR drop of up to 7.1 dB at 1 percent data), whereas pretraining consistently mitigates this deficit, yielding stable performance even under limited data availability. Beyond 50 percent data, the performance gap narrows, underscoring pretraining's importance in resource-constrained scenarios and its role in enabling large-scale ECG view synthesis.

Table 9: Data-efficiency analysis on the CPSC2018 dataset.

| Data volume | 1% (44) | | 5% (223) | | 10% (446) | | 50% (2234) | | 100% (4468) | |
|---|---|---|---|---|---|---|---|---|---|---|
| Pretrain | NO | YES | NO | YES | NO | YES | NO | YES | NO | YES |
| PSNR | 24.68 | 31.75 | 29.04 | 31.91 | 30.05 | 31.89 | 31.07 | 32.14 | 31.67 | 32.06 |
| SSIM | 0.936 | 0.978 | 0.961 | 0.978 | 0.974 | 0.979 | 0.975 | 0.979 | 0.976 | 0.979 |

### E.2 VALIDATING ON-THE-FLY CALIBRATION UNDER LEAD DEVIATIONS

One of the primary objectives of On-the-fly Calibration is to perform individual-specific calibration of ECG signals by compensating for deviations in relative lead angles arising from two key sources: (1) variability in electrode placement introduced during manual clinical setup, and (2) inter-individual anatomical differences (*e.g.*, variations in heart position and thoracic structure), as illustrated in Fig. 9.

To evaluate the effectiveness of our learnable angular correction parameter $(d\theta, d\varphi)$, we introduce 10°, 20°, and 30° angular deviation into the CPSC2018 dataset inputs and compare the result before and after the On-the-fly Calibration stage. As shown in Table 10, the quality of the result signals drop larger progressively in the absence of correction. Once the model applied $(d\theta, d\varphi)$ and On-the-fly Calibration stage, both PSNR and SSIM returned to values close to the unperturbed baseline, demonstrating that our calibration mechanism effectively compensates for electrode misplacement and anatomical variability.

To achieve individualized model calibration, NEF-NET V2 employs an Angle Embed block to encode angular information and a Geometric View Transformer block to perform view transformation. Through adaptive fine-tuning of these modules, On-the-fly Calibration aligns recorded ECG views with their anatomically consistent orientations, thereby cor-

Table 10: The impact of On-the-fly Calibration on CPSC2018 (Dataset)

| Deviation | Uncorrected | | After correction | |
|---|---|---|---|---|
| | PSNR | SSIM | PSNR | SSIM |
| 0 | 32.08 | 0.979 | – | – |
| 10 | 30.71 | 0.971 | 33.24 | 0.983 |
| 20 | 28.79 | 0.965 | 33.09 | 0.982 |
| 30 | 26.53 | 0.960 | 33.09 | 0.982 |

recting electrode placement errors and accounting for subject-specific anatomical variability. This calibration step ensures that NEF-NET V2 can effectively adapt to individual physiological and acquisition-related differences, ultimately improving its ability to synthesize accurate ECG panoramas across diverse patient populations.

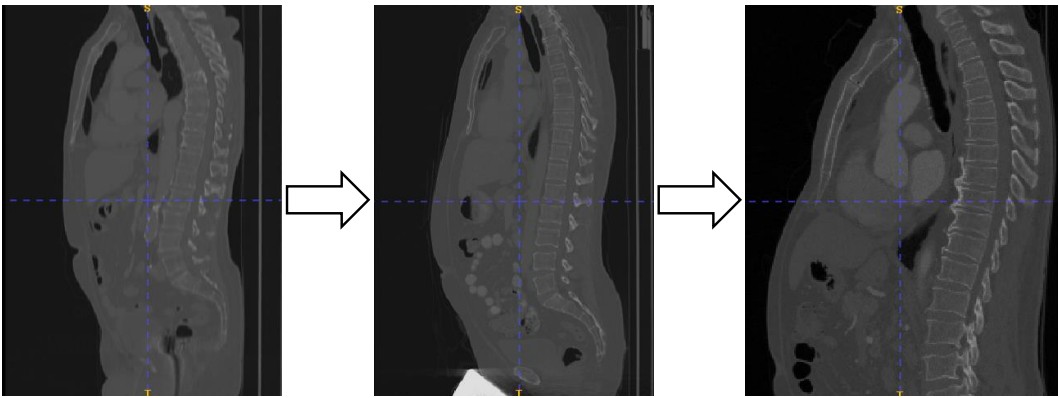

Figure 9: Illustration of inter-individual anatomical variability

### E.3 ABLATION OF MODEL STRUCTURE

Based on the benchmark NEF-NET V2 (denoted as (A) in Table 11), which averages multi-view ECG signals and employs an encoder-decoder architecture to reconstruct the query view via an estimated electric field, we investigate the impact of our proposed components.

Specifically, (B) incorporates the *Geometric View Transformer* (GeoVT), which explicitly models spatial relationships among ECG views to progressively synthesize the query view, but lacks query-guided encoding. (C) further integrates the *View Encoder* (VEncoder), yielding NEF-NET V2, where query-view angles guide feature extraction: the query embedding functions as an angle-dependent gate that amplifies features aligned with query view while suppressing misaligned ones, thereby enhancing cross-view alignment.

Results show that GeoVT alone improves synthesis PSNR from 26.79 dB to 29.54 dB, demonstrating the effectiveness of explicit geometric modeling. Adding VEncoder yields a substantial further gain (PSNR: 31.19 dB), validating the importance of query-aware feature encoding for precise perspective transformation. From a geometric learning perspective, GeoVT captures inter-view spatial dependencies by progressively aggregating view-consistent features, while VEncoder leverages query-angle embeddings to constrain feature extraction within the correct anatomical frame of reference. Their synergy ensures that synthesized signals remain anatomically consistent and view-coherent, even under significant electrode or anatomical variability.

Finally, (D) removes the noise perturbation $\epsilon$ and shows a slight performance drop compared to (C), confirming the stabilizing effect of noise injection during training. Overall, these results highlight the complementary contributions of GeoVT, VEncoder, and controlled noise perturbation in improving ECG view synthesis and transformation.

### E.4 ROBUSTNESS TO COMMON ECG NOISE

NEF-NET V2 is intrinsically robust to common ECG distortions such as *powerline interference* and *baseline wander*. However, signals with strong motion artifacts may significantly degrade synthesis quality. This is expected because motion artifacts are nonstationary and abrupt disturbances whose frequency components heavily overlap with those of the ECG itself, making them fundamentally difficult to remove and a long-standing challenge even in the signal-processing literature. Motion artifacts typically exhibit amplitudes far exceeding those of the underlying ECG signal, effectively rendering the data unusable. Therefore, segments with severe artifacts are often discarded during data preprocessing. Moreover, mitigating such artifacts is primarily achieved through optimized

Table 11: Ablation study on CPSC2018 dataset (lead configuration: 3,8,1) evaluated in the Device Calibration stage.

| | $\epsilon$ | Components GeoVT | VEncoder | Synthesis PSNR | SSIM | Reconstruction PSNR | SSIM |
|---|---|---|---|---|---|---|---|
| A | ✓ | – | – | 26.79 | 0.958 | 28.47 | 0.960 |
| B | ✓ | ✓ | – | 29.54 | 0.972 | 32.22 | 0.971 |
| C | ✓ | ✓ | ✓ | **31.19** | **0.976** | **35.79** | **0.981** |
| D | – | ✓ | ✓ | 30.04 | 0.976 | 35.41 | 0.976 |

circuit-level design during signal acquisition, as algorithmic post-processing alone is often inadequate for their effective removal (Lee et al., 2023; Torfs et al., 2014).

We first compared the synthesis performance on signals containing powerline interference and baseline wander against that on clean signals. Visual inspection shows that the generated outputs effectively suppress these noise components. However, because the synthesized signals are clean while the reference signals remain noisy, the direct comparison leads to lower raw PSNR/SSIM values. To ensure a fair evaluation, we therefore denoised the reference signals and repeated the tests. The results are presented in the following table 12.

Overall, noise exerts only a limited influence on the model. Since the generator tends to produce clean signals, discrepancies arise when the reference contains noise, leading to lower raw metrics even when the synthesized outputs are visually superior.

Table 12: Synthesis performance (PSNR/SSIM) with noisy reference signals.

| Noise Type | ChinaDB | CPSC2018 | Tianchi | PTB-XL |
|---|---|---|---|---|
| | | *Before Filtering* | | |
| Powerline interference | 32.14 / 0.979 | 33.11 / 0.983 | 34.57 / 0.980 | 33.04 / 0.981 |
| Baseline wander | 32.53 / 0.982 | 33.47 / 0.983 | 34.69 / 0.981 | 33.25 / 0.981 |
| Clear signal | 33.24 / 0.986 | 34.18 / 0.987 | 35.14 / 0.987 | 34.17 / 0.983 |
| | | *After Filtering* | | |
| Powerline interference | 33.21 / 0.985 | 34.09 / 0.985 | 35.12 / 0.987 | 34.04 / 0.982 |
| Baseline wander | 33.08 / 0.984 | 34.13 / 0.985 | 34.97 / 0.986 | 33.95 / 0.982 |
| Clear signal | 33.24 / 0.986 | 34.18 / 0.987 | 35.14 / 0.987 | 34.17 / 0.983 |

### E.5 CLINICAL EVALUATION

The ultimate value of synthesized ECG signals lies in their diagnostic fidelity, particularly the ability to preserve pathological features across different viewing angles. In this section, we analyze the synthesized views for pathological cases of LBBB and RBBB.

As shown in the figure 10. Across the core diagnostic leads (V1 and V6), the panoramic ECG signals generated by NEF-NET V2 exhibit a high degree of consistency with the corresponding real recordings. In the LBBB case, the synthesized V1 lead ($\theta = 90°$, $\phi = -10°$) accurately reproduces the characteristic wide and deep QS/rS morphology. Likewise, the synthesized V6 lead ($\theta = 96°$, $\phi = 90°$) clearly presents the broad, notched R wave accompanied by the expected secondary ST–T changes. In the RBBB case, the synthesized V1 lead precisely captures the hallmark rsR′ "rabbit-ear" pattern, reflecting the delayed right ventricular depolarization vector. These observations demonstrate that the panoramic ECG model effectively captures the pathophysiological signatures of bundle branch block and is capable of reproducing diagnostically decisive waveform features at standard lead locations.

As shown in Fig. 11, NEF-NET V2 demonstrates strong capability in generating ECGs from nonstandard, virtual views. Given three sets of spherical coordinates: ($\theta = 90°, \phi = 130°$),

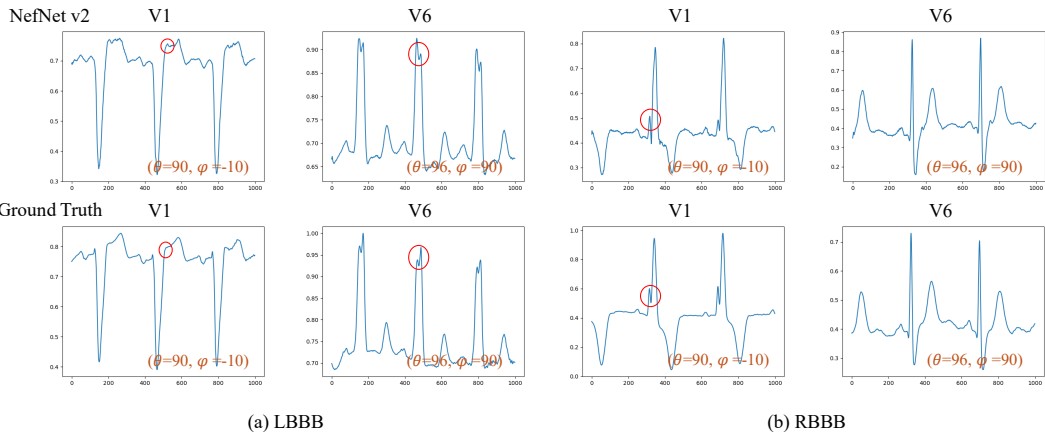

Figure 10: ECG recordings from leads V1 and V6 of patients with LBBB and RBBB, along with the corresponding V1 and V6 signals synthesized by NEF-NET V2 from leads I, II, and V5.

$(\theta = 132°, \phi = -99°)$, and $(\theta = 140°, \phi = 100°)$, NEF-NET V2 produces the corresponding synthesized views.

The results indicate that the synthesized signals consistently preserve the essential pathological characteristics across different viewpoints. For RBBB, the virtual view at $(\theta = 132°, \phi = -99°)$ captures the typical rsR′ morphology. For LBBB, the virtual view at $(\theta = 90°, \phi = 130°)$ reveals the broad, notched R wave reflecting dominant left ventricular depolarization, while the view at $(\theta = 132°, \phi = -99°)$ presents a wide and deep QS pattern. These findings show that NEF-NET V2 does not merely map waveforms but effectively reconstructs the three-dimensional cardiac depolarization vector, enabling the generation of ECG signals as if observed from any point on the cardiac sphere and thereby overcoming the physical constraints of conventional electrode placement.

Based on this principle, when diagnosing cardiovascular diseases that rely on specific lead positions, NEF-NET V2 offers a distinct advantage. For example, in the diagnosis of Brugada syndrome, the characteristic ECG pattern may not appear in the standard V1–V2 leads recorded at the fourth intercostal space. Clinically, the electrodes must be repositioned to the second or third intercostal space to significantly improve detection sensitivity. In contrast, NEF-NET V2 can directly generate the corresponding virtual-view ECG simply by specifying the desired viewpoint coordinates, thereby providing clinicians with an optimized observation window without physically repositioning the electrodes.

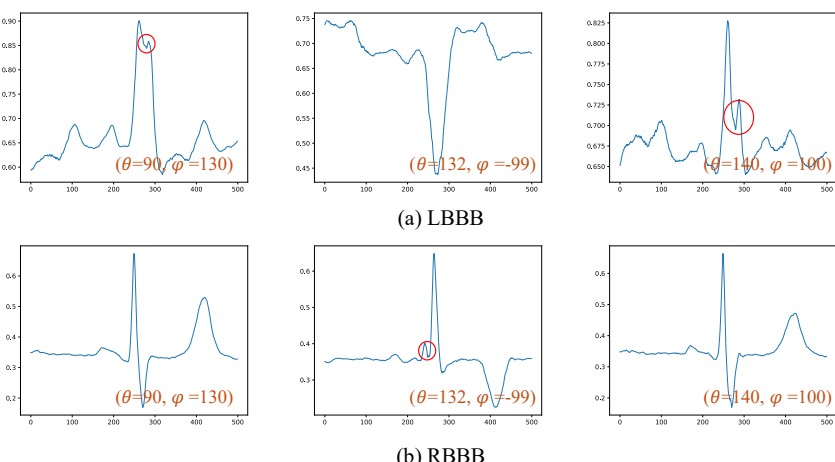

Figure 11: Arbitrary-view ECG signals of LBBB and RBBB patients synthesized by NEF-NET V2 from leads I, II, and V5.

Since the pathological signatures of LBBB and RBBB manifest most prominently in leads V1 and V6, we asked two independent board-certified cardiologists to evaluate the diagnostic quality of the generated V1 and V6 signals under these two conditions. Each waveform was scored on a 0–5 scale (5 = excellent fidelity, 3 = clinically acceptable, 0 = misleading interpretation). Across 60 evaluations per cardiologist, all scores ranged between 3 and 5.

These evaluations show that NEF-NET V2 faithfully preserves the diagnostic morphology of bundle branch blocks, including the wide QS/rS complexes in V1 and broad notched R waves in V6 for LBBB, and the characteristic rsR pattern in V1 for RBBB. Importantly, neither expert identified any clinically misleading artifacts. The consistently high ratings across both experts confirm that the synthesized signals maintain clinically meaningful fidelity, further supporting the utility of NEF-NET V2 for pathological ECG synthesis.

Table 13: Cardiologist evaluation of synthesized V1/V6 signals for LBBB and RBBB. Scores range from 0–5 (higher is better).

| Diagnosis | 5 | 4 | 3 | 2 | 1 | 0 | Total |
|---|---|---|---|---|---|---|---|
| *Cardiologist 1* | | | | | | | |
| LBBB | 25 | 5 | 0 | 0 | 0 | 0 | 30 |
| RBBB | 28 | 2 | 0 | 0 | 0 | 0 | 30 |
| *Cardiologist 2* | | | | | | | |
| LBBB | 25 | 3 | 2 | 0 | 0 | 0 | 30 |
| RBBB | 29 | 0 | 1 | 0 | 0 | 0 | 30 |

For the panoramic dataset Panobench, we further provide a visual comparison between the real panoramic ECG (Fig. 4) and the panorama synthesized by NEF-NET V2 (Fig. 12). In Fig. 12, orange dashed boxes indicate the recorded input views (real signals), blue dashed boxes highlight the reconstructed views corresponding to supervised targets, and all remaining unboxed views represent fully synthesized signals. A close inspection reveals that the reconstructed views preserve finer morphological details due to direct supervision. In addition the generated views appear slightly smoother—an expected behavior of generative models. Nonetheless, the generated signals remain highly consistent with the real panoramic ECG in Fig. 4, capturing the global morphology and inter-lead relationships across the full 48-view panorama.

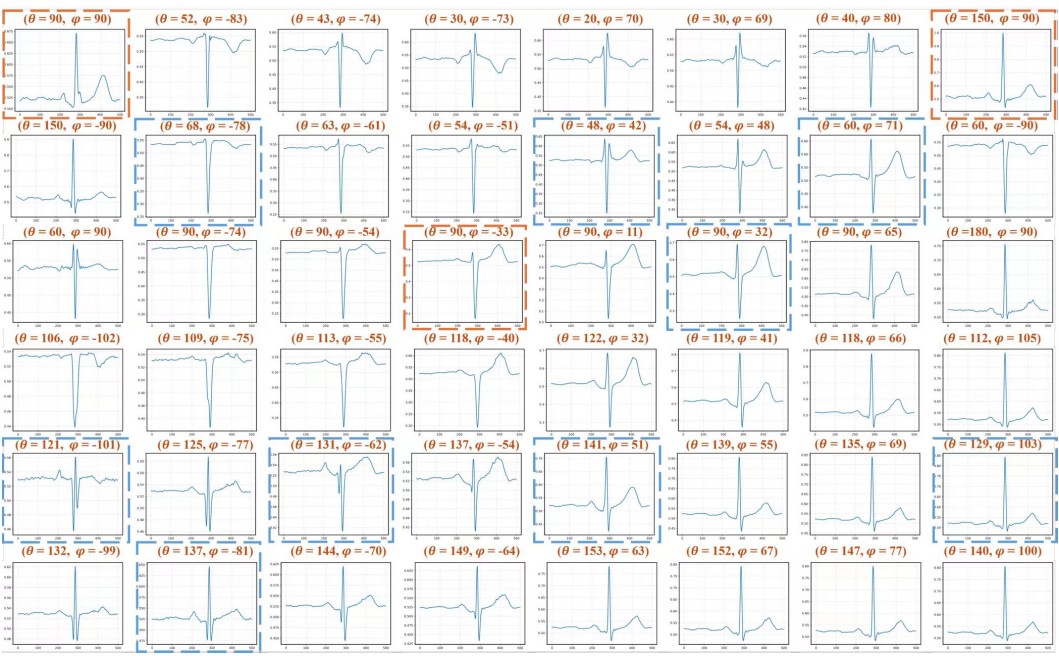

Figure 12: Representative example synthesized by NEF-NET V2 using Panobench. For intuitive comparison with the ground-truth panoramic ECG in Fig. 12, orange dashed boxes indicate the recorded ECG views, blue dashed boxes denote the views used as supervision during training, and all remaining views are synthesized by NEF-NET V2.

