# OpenReview forum: "Nef-Net v2: Adapting Electrocardio Panorama in the wild"
_ICLR.cc/2026/Conference — ICLR 2026 Poster_

### Official Review · Reviewer_yWqz · 2025-10-29

**Soundness:** 3
**Presentation:** 3
**Contribution:** 3
**Rating:** 6
**Confidence:** 4

**Summary:**

The paper proposes NEF-NET+, a geometry-aware ECG transformation model that replaces field reconstruction with a direct view-to-view learning strategy.
It introduces a multi-stage adaptation pipeline to improve real-world robustness.
A new 48-view dataset, Panobench, is constructed for spatially precise evaluation of cross-lead ECG reconstruction.

**Strengths:**

- The combination of direct view transformation and geometry-aware attention effectively captures spatial lead dependencies while simplifying the modeling pipeline.
- The three-stage calibration framework is well-aligned with real-world ECG acquisition and device variability.
- The proposed 48-view Panobench dataset expands the spatial evaluation scope beyond traditional 12-lead ECGs

**Weaknesses:**

- Limited Validation of Geometry-Aware Attention: While the MGAA and GeoVT modules are claimed to model inter-lead angular relationships, the paper lacks quantitative evidence that these mechanisms genuinely capture geometric or biophysical correspondence.
While the geometry-aware mechanism is formally defined and integrated into the architecture, the paper does not present attention-weight visualizations nor any correlation analysis between learned attention weights and actual spatial distances or electric potential gradients.

- Lack of Clinical Interpretability: The performance evaluation relies heavily on signal reconstruction metrics, which do not directly reflect diagnostic or physiological relevance.
The absence of task-oriented evaluation metrics limits the interpretability and translational value of the proposed framework in clinical contexts.

- Computational Efficiency and Deployment Feasibility: Although the architecture simplifies the overall transformation pipeline, the paper does not quantify the practical cost of on-the-fly calibration, which requires per-examination adaptation.
The manuscript does not report time-to-adapt (wall-clock latency from raw input to an adapted model), compute/memory budget for the calibration step on realistic hardware, or convergence stability under streaming/telemetry conditions.
This omission makes it difficult to assess whether the per-patient adaptation loop can meet real-time throughput constraints in clinical workflows and whether maintaining adapted parameters across encounters is feasible on resource-constrained medical devices.

- Limited Accessibility of the Panobench Dataset: While the appendix provides detailed descriptions of the Panobench dataset and its construction process, the dataset itself is not publicly available at the time of writing.
Given that Panobench serves as a central contribution and evaluation benchmark, the lack of open access limits reproducibility and independent verification of the reported results (the paper only states that code and data “will also be released publicly”).

**Questions:**

It would be helpful to address the weaknesses if possible.

**Details Of Ethics Concerns:**

There is a need to explicitly document and verify compliance with IRB approval, informed consent procedures, and data governance (de-identification, re-identification risk, and release policy) for the human-subject ECG/CT data used in this study.

---

> ### Author Response · Authors · 2025-11-19
> **To Reviewer yWqz (Part I)**
>
> We appreciate the reviewer’s recognition that our direct view transformation with geometry-aware attention effectively captures spatial lead dependencies while simplifying the modeling pipeline. We also thank the reviewer for noting that our three-stage calibration framework aligns well with real-world ECG acquisition and device variability, and for acknowledging that our 48-view Panobench greatly expands the spatial evaluation scope beyond the traditional 12-lead setting.Below we provide a point-by-point response to your questions and hope it clarifies all of your concerns.
>
> ### **Q1. Insufficient Validation of Geometry-Aware Attention Mechanisms**
>
> **Response:**
>
> To address this concern, we provide quantitative evidence that the MGAA and GeoVT modules indeed learn anatomically meaningful inter-lead relationships. As Panobench contains 48 views, we focus the analysis on the standard 12-lead ECG for clarity.
>
> Below we visualize the normalized attention contributions for generation V1–V6 under different input-lead combinations.
>
> #### **(1) Inputs: Lead I, Lead II, V1 → Generating V2–V6**
> |        | V2   | V3   | V4   | V5   | V6   |
> |--------|------|------|------|------|------|
> | Lead I |0.21 | 0.37 | 0.36 | 0.35 | 0.35 |
> | Lead II|0.15 | 0.29 | 0.33 | 0.37 | 0.43 |
> | V1     |0.64 | 0.34 | 0.28 | 0.28 | 0.22 |
>
> #### **(2) Inputs: Lead I, Lead II, V3 → Generating V1, V2, V4, V5, V6**
> |        | V1   | V2   | V4   | V5   | V6   |
> |--------|------|------|------|------|------|
> | Lead I | 0.48 | 0.29 |0.25 | 0.28 | 0.33 |
> | Lead II| 0.12 | 0.15 |0.15 | 0.29 | 0.36 |
> | V3     | 0.40 | 0.55 |0.60 | 0.43 | 0.31 |
>
> #### **(3) Inputs: Lead I, Lead II, V5 → Generating V1–V4, V6**
> |        | V1   | V2   | V3   | V4   |V6   |
> |--------|------|------|------|------|------|
> | Lead I | 0.41 | 0.37 | 0.35 | 0.14 |0.13 |
> | Lead II| 0.29 | 0.29 | 0.19 | 0.13 |0.12 |
> | V5     | 0.30 | 0.34 | 0.46 | 0.73 |0.75 |
>
> The attention weights exhibit a strong positive correlation with the geometric proximity between the synthesized and target views. This behavior aligns exactly with the known physical chest-lead geometry.
>
> We further computed pairwise Pearson correlations among chest leads based on their corresponding ECG signals (on CPSC 2018 testset):
>
> |  | V1   | V2   | V3   | V4   | V5   | V6   |
> |-|-|--|-|-|-|-|
> | V1     | 1    | 0.87 | 0.45 | 0.15 | 0.11 | 0.11 |
> | V2     | 0.87 | 1    | 0.76 | 0.43 | 0.39 | 0.38 |
> | V3     | 0.45 | 0.76 | 1    | 0.88 | 0.84 | 0.83 |
> | V4     | 0.15 | 0.43 | 0.88 | 1    | 0.99 | 0.98 |
> | V5     | 0.11 | 0.39 | 0.84 | 0.99 | 1    | 0.99 |
> | V6     | 0.11 | 0.38 | 0.83 | 0.98 | 0.99 | 1    |
>
> One can see that, across all settings, the learned attention patterns follow the **actual geometric relationships** between chest leads:
> - The model assigns **highest attention to the anatomically closest available lead**.
> - Attention weights **decay smoothly** with increasing inter-lead anatomical distance.
> - The pattern is consistent whether the mid-chest (V3) or lateral leads (V5) are provided as inputs.
>
> These results demonstrate that the geometry-aware modules are indeed learning **biophysically meaningful, spatially structured** inter-lead relationships.

---

> ### Author Response · Authors · 2025-11-19
> **To Reviewer yWqz (Part II)**
>
> ### **Q2. Lack of Clinical Interpretability**
>
> **Response:**
>
> We agree that clinical interpretability is essential and that evaluating only reconstruction metrics is insufficient. The goal of our framework is to generate panoramic ECGs that retain *diagnostic morphology* across diverse acquisition settings.
>
> Since clinicians interpret ECGs primarily through waveform morphology rather than point-wise amplitudes, SSIM is used to assess structural similarity, while diagnostic relevance is evaluated separately in *Section 4: Performances on Different Diseases*: Table 4 shows that Nef-Net+ synthesizes morphology-consistent signals under abnormal conditions. In addtion, in *Figure 5*, the three-stage pipeline correctly reconstructs clinically meaningful features, e.g., in samples 2300–2800, the synthesized T-wave amplitude becomes lower than the R-wave, matching the true diagnostic pattern.
>
> We will further strengthen this validation with two additional experiments:
>
> - View-specific pathological reconstruction: We include synthesized panoramic signals for left and right bundle branch block, both of which exhibit distinct and view-dependent waveforms. The results show that these characteristic conduction patterns are preserved.
>
> - Downstream diagnostic evaluation and Cardiologist review: We train an 1D classifier (following reference [1]) as a proxy for clinical assessment on some common heart diseases (AF, LBBB, RBBB, PAC) and consult two cardiologists to evaluate the pathological validity of our generated signals on 30 cases.
>
> #### **Diagnostic classifier performance**
> |      | AF   | LBBB | RBBB | PAC
> |-----|--|--|--|------|
> | Three-leads       | 0.81 | 0.71 | 0.72 | 0.58
> | Generated 12-lead | 0.82 | 0.77 | 0.79 | 0.68
> | Real 12-lead   | 0.84 | 0.78 | 0.83 | 0.71
>
> **Our results show that the model captures diseased patterns with high fidelity.**
>
> The synthesized signals for LBBB and RBBB conditions in key leads (V1 and V6) were evaluated by two independent expert cardiologists. The diagnostic quality of characteristic waveforms was rated on a 0-5 scale (5 is better). All scores fell within the range of 3 to 5, indicating clinically acceptable to excellent feature fidelity.
>
> #### **Cardiologist 1 evaluation**
> | Diagnosis | 5 | 4 | 3| 2 | 1 | 0 | Total |
> |--|---|--|---|---|---|---|---|
> | LBBB      | 25  | 5  | 0 | 0  |0 |0  | 30  |
> | RBBB      | 28 | 2  | 0  |0  |0 |0 | 30  |
>
> #### **Cardiologist 2 evaluation**
> | Diagnosis | 5 | 4 | 3 | 2 | 1 | 0 | Total |
> |--|---|--|---|---|---|---|---|
> | LBBB      | 25          | 3    | 2     | 0 | 0 | 0  | 30    |
> | RBBB      | 29          | 0    | 1   | 0 | 0 | 0      | 30   |
>
> These evaluations collectively show that our model not only preserves high structural similarity but also maintains *diagnostic integrity* across multiple abnormal conditions. We have included these results in the revised version.
>
> [1] Francia P, Balla C, Paneni F, et al. Left bundle‐branch block—pathophysiology, prognosis, and clinical management. Clinical Cardiology: An International Indexed and Peer‐Reviewed Journal for Advances in the Treatment of Cardiovascular Disease, 2007.
>
> [2] Ikeda T. Right bundle branch block: current considerations. Current Cardiology Reviews, 2021.
>
> [3] Jie Hu, Li Shen, Gang Sun, Squeeze-and-excitation networks, CVPR, 2018.

---

> ### Author Response · Authors · 2025-11-19
> **To Reviewer yWqz (Part III)**
>
> ### **Q3. Computational Efficiency and Deployment Feasibility**
>
> **Response:**
>
> Nef-Net+ is a lightweight view-transformation model with only **3.8M parameters (15.2 MB, FP32)**. Because it avoids explicit electric-field reconstruction and performs only geometry-aware view transformation, the on-the-fly calibration stage updates **only angle-related parameters (<6% of the model)**. This design keeps both compute and memory usage very modest. The model was fine-tuned for 100 iterations (Appendix D.3) using an NVIDIA GeForce RTX 2080 Ti GPU. To provide a concrete estimate of the calibration time, we evaluated different iteration counts and observed a clear trade-off between calibration accuracy and computational cost, as summarized in the table below.
>
> | Iterations | Time (s) | PSNR |
> |-----------|----------|-------|
> | 10        | 0.72     | 33.11 |
> | **100**   | **6.34** | **33.62** |
> | 200       | 13.19    | 33.59 |
> | 1000      | 62.82    | 32.87 |
>
> Our experiments show that 100 iterations achieve the best calibration performance which is within an acceptable range. In addition, because clinical ECG recordings are typically 10 seconds long, the 5-second calibration window overlaps substantially with the acquisition of the remaining 5 seconds. As a result, the calibration time of 6.34 seconds introduces minimal additional latency in practical use. In summary, these results indicate that on-device calibration is feasible even on resource-constrained medical hardware.
>
> ### **Q4. Limited Accessibility of the Panobench Dataset**
>
> **Response:**
>
> To support reproducibility during the review phase, the supplementary materials already provide the full experimental code and a complete 10-second Panobench sample with raw signals and metadata, enabling reviewers to reproduce the entire processing and modeling pipeline.
>
> Panobench required more than 18 months of continuous engineering and data collection with a specialized 48-view acquisition system. Its public release must therefore be coordinated carefully to ensure proper documentation, licensing, and long-term maintenance.
>
> We reaffirm our commitment that the full Panobench dataset and our codes will be made publicly available immediately upon acceptance, ensuring full reproducibility and independent verification.
>
> We welcome the AC and reviewers to oversee this process.

---

### Official Review · Reviewer_MVub · 2025-10-29

**Soundness:** 2
**Presentation:** 2
**Contribution:** 2
**Rating:** 2
**Confidence:** 4

**Summary:**

The paper introduces Nef-Net+, an enhanced framework for panoramic ECG synthesis. The model supports arbitrary-length signal generation from any desired view, aims to generalize across different ECG devices, and compensates for operator-induced deviations in electrode placement. To evaluate panoramic ECG synthesis, the authors also present Panobench, a new benchmark dataset including 9,369 recordings with 48 views per subject, designed to capture the spatial variability of cardiac electrical activity.

**Strengths:**

1) The paper addresses an important practical limitation of ECG modeling, i.e., the restricted number of observation viewpoints, and proposes an architectural improvement over Nef-Net.

2) The Panobench dataset could be of interest to the biomedical ML community.

**Weaknesses:**

1) **The paper does not align with the conference scope**. The contribution is highly domain-specific and would be more appropriate for an applied or benchmark-oriented venueApplied/Benchmark track conference.

2) Several paragraphs are written in a dense and overly technical style (e.g., 068-071, 152-155). A clearer presentation of the original Nef-Net architecture would greatly help to understand the improvements introduced in Nef-Net+, since the latter builds directly upon the former. For instance, the description of “direct view-to-view transformation” versus “neural electrocardio field reconstruction” (lines 140-142) it is important but not explained.

3) Weak evaluation (see *Questions*).

4) The discussion of related work on ECG reconstruction is limited and overlooks several recent and top-tier contributions in this area, to cite a few:

[1] Alex Lence, Federica Granese, Ahmad Fall, Blaise Hanczar, Joe-Elie Salem, Jean-Daniel Zucker, Edi Prifti: ECGrecover: A Deep Learning Approach for Electrocardiogram Signal Completion. KDD (1) 2025: 2359-2370

[2] Juan Miguel Lopez Alcaraz, Nils Strodthoff: Diffusion-based Time Series Imputation and Forecasting with Structured State Space Models. Trans. Mach. Learn. Res. 2023 (2023)

[3] Jinho Joo, Gihun Joo, Yeji Kim, Moo-Nyun Jin, Junbeom Park, Hyeonseung Im: Twelve-Lead ECG Reconstruction from Single-Lead Signals Using Generative Adversarial Networks. MICCAI (7) 2023: 184-194

**Questions:**

1) Could the authors clarify what exactly *device heterogeneity* refers to in their experiments? It looks like by heterogeneity, they mean data from different datasets.

2) Was the Panobench dataset curated or validated by medical professionals?

3) How were the ECG datasets preprocessed before training? The paper combines recordings with different sampling frequencies (1000/500/250 Hz) and signal durations. Details regarding signal preprocessing are fundamental but completely missing from the paper.

4) What is the rationale for using SSIM as an evaluation metric? Standard metrics exist for ECG synthesis and reconstruction (e.g., see [1, 2, 3]). A morphological analysis of ECG features such as peaks, waves, and segments would also be necessary to assess clinical fidelity. In addition, no statistical information (e.g., standard deviations) is reported for the quantitative results.

5) It is unclear whether the experiments are conducted at the heartbeat level (as in the original Nef-Net) or on continuous ECG recordings. If the latter is the case, it becomes unclear how a fair comparison with Nef-Net is ensured, since that model was evaluated on single-beat reconstructions.

6) The paper repeatedly states that Nef-Net+ supports *arbitrary-length ECG synthesis*. How is this implemented in practice? Are sequences chunked, streamed, or processed recurrently?

---

> ### Author Response · Authors · 2025-11-18
> **To Reviewer MVub (Part I)**
>
> We appreciate the reviewer’s acknowledgment that the paper tackles an important practical limitation in ECG modeling and introduces a meaningful architectural improvement over Nef-Net. We are also grateful that the reviewer found Panobench to be of interest to the biomedical ML community, which we believe highlights the broader relevance of our contributions to real-world applications in healthcare and computational biology. Below we provide a point-by-point response to your questions and hope it clarifies all of your concerns.
>
> ### **Q1. Concern: The paper does not align with the ICLR conference scope**
>
> **Response:**
>
> We would like to respectfully clarify that our work is fully aligned with ICLR’s stated scope. The official 2026 *Call for Papers* explicitly highlights “applications in ... healthcare, biology, sustainability ... ” as a core area of interest. ECG modeling is an important task in healthcare and biology, and therefore fits naturally within this scope.
>
> ICLR has also consistently accepted deep learning research focused on ECG analysis. For example, ICLR 2025 features some ECG-related papers:
>
> > Reading Your Heart: Learning ECG Words and Sentences via Pre-training ECG Language Model
>
> > Guiding Masked Representation Learning to Capture Spatio-Temporal Relationship of Electrocardiogram
>
> These precedents demonstrate that ECG modeling is regarded as a mainstream and appropriate research direction for ICLR.
>
> More broadly, AI models across all domains are ultimately developed to support human well-being. Our work introduces a real-time and robust panoramic ECG visualization approach together with Panobench, a valuable benchmark for the community, which contributes directly to this mission.
>
> ### **Q2. Dense writing and unclear distinction between “direct view-to-view transformation” and “neural electrocardio field reconstruction”**
>
> **Response:**
>
> Some parts of the manuscript are indeed compact due to page-limit constraints, and we have clarified these descriptions in the revision. Below is a brief summary of the distinctions:
>
> **(1) Direct view-to-view transformation (Nef-Net+):**
> This is a *pairwise* deterministic mapping: the model converts the observed lead signals into the target lead through a single-step transformation, without modeling any shared geometric prior, such as the *“electrocardio field representation”* described in the original Nef-Net paper.
>
> **(2) Neural electrocardio field reconstruction (original Nef-Net).**
> Nef-Net+ instead models a **continuous latent cardiac field (“electrocardio field representation” as stated in the original paper)** from all leads.
> - The field encodes spatially coherent electrical activity.
> - Each target lead is produced by *querying* this latent field at the anatomical location defined by its electrode angle.

---

> ### Author Response · Authors · 2025-11-19
> **To Reviewer MVub (Part II)**
>
> ### **Q3. Weak evaluation and rationale for using SSIM; need for morphological/clinical analysis**
>
> **Response:**
>
> We would like to emphasize that Table 4 evaluates the preservation of diagnostic features, and Figure 5 illustrates how our three-stage framework captures morphological detail. Below, we would like to address your concerns in detail.
>
> **1. Why SSIM and PSNR?**
> **(1)** Why SSIM?
>
> Actually, our primary evaluation objective is **signal-level similarity**, as our framework is designed to generate panoramic ECGs for *visual inspection* by clinicians. In routine clinical practice, cardiologists make decisions primarily based on **waveform morphology** and **inter-lead relationships**, rather than exact point-wise amplitude matching.
>
> For this reason, we adopt **SSIM** as a core metric:
> - SSIM emphasizes *structural fidelity*, which aligns with how clinicians interpret ECG shapes.
> - It is less sensitive to global linear amplitude transformations caused by differing device gains, baseline variations, or preprocessing (e.g., min–max normalization).
> - **For fair comparison, the Nef-Net [1] model utilized the SSIM metric.**
>
> **(2)** Why PSNR?
> - Amplitude-based metrics, such as RMSE and MAE (employed in [2–4]), measure point-wise amplitude differences. Like PSNR, these metrics are fundamentally tied to amplitude errors, as PSNR is mathematically derived from the RMSE.
> - **For fair comparison, the Nef-Net [1] model utilized the PSNR metric.**
>
> $$
> \text{PSNR} = 20 \cdot \log_{10}\left(\frac{MAX}{\text{RMSE}}\right)
> $$
>
> **(3)** Here we also report the RMSE (↓) and MAE (↓) performances below for reference.
>
> | Method      | ChinaDB (RMSE/MAE) | CPSC2018 (RMSE/MAE) | Tianchi  (RMSE/MAE) | PTB-XL (RMSE/MAE) |
> |-------------|----------------------|------------------------|----------------------|----------------------|
> | ECGRecover [2] |  0.028/0.013         | 0.033 / 0.014         | 0.027/ 0.009       |  0.037 / 0.015        |
> | EKGAN [3] | 0.021/0.009        |  0.025/0.010         | 0.016/0.007        | 0.028/0.012         |
> | SSSD [4] | 0.026/0.011     |0.027/0.012       | 0.023/0.010              | 0.029/0.014                   |
> | **Nef-Net+** | **0.017/0.007** | **0.016/0.007**       | **0.011/0.004**     | **0.019/0.007**     |
>
> **2. Morphological and clinical evaluations**
>
> Nef-Net+ effectively preserves morphological details, as visually illustrated in Figure 5. Taking the heartbeat between samples 2300–2800 in the fourth row as an example: the ground-truth signal shows a T-wave amplitude smaller than the R-wave—a key diagnostic feature. While the outputs from the first two stages incorrectly show a larger T-wave (which would be pathologically abnormal), the final output after on-the-fly calibration accurately reproduces the correct relationship, resulting in a diagnosable-level synthesis.
>
> We will further strengthen this validation with two additional experiments:
>
> - View-specific pathological reconstruction: We include synthesized panoramic signals for left and right bundle branch block, both of which exhibit distinct and view-dependent waveforms. The results show that these characteristic conduction patterns are preserved.
>
> - Downstream diagnostic evaluation and Cardiologist review: We train an 1D classifier (following the structure as in [5]) as a proxy for clinical assessment on some common heart diseases (AF, LBBB, RBBB, PAC) and consult two cardiologists to evaluate the pathological validity of our generated signals on 30 cases.
>
> #### **Diagnostic classifier performance**
> |   | AF   | LBBB | RBBB | PAC
> |-|--|--|--|--|
> | 3-leads       | 0.81 | 0.71 | 0.72 | 0.58
> | Generated 12-lead | 0.82 | 0.77 | 0.79 | 0.68
> | Real 12-lead   | 0.84 | 0.78 | 0.83 | 0.71
>
> **Our results show that the model captures diseased patterns with high fidelity.**
>
> The synthesized signals for LBBB and RBBB conditions in key leads (V1 and V6) were evaluated by two independent expert cardiologists. The diagnostic quality of characteristic waveforms was rated on a 0-5 scale (5 is better). All scores fell within the range of 3 to 5, indicating clinically acceptable to excellent feature fidelity.
>
>
> [1] Chen J, Zheng X, Yu H, et al. Electrocardio panorama: synthesizing new ECG views with self-supervision[J]. arXiv preprint arXiv:2105.06293, 2021.
>
> [2] Alex Lence, Federica Granese, Ahmad Fall, Blaise Hanczar, Joe-Elie Salem, Jean-Daniel Zucker, Edi Prifti: ECGrecover: A Deep Learning Approach for Electrocardiogram Signal Completion. KDD (1) 2025: 2359-2370
>
> [3] Jinho Joo, Gihun Joo, Yeji Kim, Moo-Nyun Jin, Junbeom Park, Hyeonseung Im: Twelve-Lead ECG Reconstruction from Single-Lead Signals Using Generative Adversarial Networks. MICCAI (7) 2023: 184-194
>
> [4] Juan Miguel Lopez Alcaraz, Nils Strodthoff: Diffusion-based Time Series Imputation and Forecasting with Structured State Space Models. Trans. Mach. Learn. Res. 2023 (2023)
>
> [5] Jie Hu, Li Shen, Gang Sun, Squeeze-and-excitation networks, CVPR2018.
>
> *（continued below）*

---

> ### Author Response · Authors · 2025-11-19
> **To Reviewer MVub (Part III)**
>
> #### **Cardiologist 1 evaluation**
> | Diagnosis | 5 | 4 | 3| 2 | 1 | 0 | Total |
> |--|---|--|---|---|---|---|---|
> | LBBB      | 25  | 5  | 0 | 0  |0 |0  | 30  |
> | RBBB      | 28 | 2  | 0  |0  |0 |0 | 30  |
>
> #### **Cardiologist 2 evaluation**
> | Diagnosis | 5 | 4 | 3 | 2 | 1 | 0 | Total |
> |--|---|--|---|---|---|---|---|
> | LBBB      | 25          | 3    | 2     | 0 | 0 | 0  | 30    |
> | RBBB      | 29          | 0    | 1   | 0 | 0 | 0      | 30   |
>
> These evaluations collectively show that our model not only preserves high structural similarity but also maintains *diagnostic integrity* across multiple abnormal conditions.  We have included these results in the revised version.
>
> **3. Statistical information** (for Q3)
>
> For completeness, **all experiments were repeated multiple times**, and the resulting standard deviations were consistently very small. Because these fluctuations did not affect any conclusions, we omitted them for readability, but we have included these results in the revised version.
>
> ### **Q4. Limited coverage of recent ECG reconstruction literature**
>
> We sincerely thank the reviewer for highlighting these strong and relevant works. However, our original submission did not include comparisons with them because we address different tasks.
>
> Our primary objective is **ECG synthesis from arbitrary view angles**, which differs fundamentally from the **lead-reconstruction** setting addressed in [1–3]. In our manuscript, reconstruction is included only as a supporting experiment to benchmark synthesis quality, rather than as the main task. For this reason, we did not originally position our work as directly comparable to these reconstruction-focused methods.
>
> Here we further compare with these three works in signal reconstruction (3 to 9 setting). The results are summarized below.
>
> #### **Reconstruction performance comparison (PSNR / SSIM)**
>
> | Method      | ChinaDB (PSNR/SSIM) | CPSC2018 (PSNR/SSIM) | Tianchi (PSNR/SSIM) | PTB-XL (PSNR/SSIM) |
> |--|----|--|--|--|
> | ECGRecover [1] | 30.47 / 0.958        | 30.12 / 0.966          | 31.47 / 0.971        | 28.57 / 0.942        |
> | EKGAN [2] | 32.76 / 0.967        | 33.35 / 0.975          | 34.39 / 0.977        | 31.71 / 0.972        |
> | SSSD [3] | 32.53/0.966    |32.67/0.972         | 33.59/0.975             | 31.42/0.972                   |
> | **Nef-Net**     | 29.59 / 0.961        | 29.12 / 0.958          | 31.44 / 0.965        | 30.22 / 0.962        |
> | **Nef-Net+** | **35.84 / 0.977** | **36.12 / 0.981**       | **37.13 / 0.982**     | **35.21 / 0.974**     |
>
> These results show that while our framework is not designed specifically for lead-reconstruction, **Nef-Net+ still achieves substantially stronger performance** than all reconstruction-focused baselines across multiple datasets. We have included these results in the revised version.
>
> [1] Alex Lence, Federica Granese, Ahmad Fall, Blaise Hanczar, Joe-Elie Salem, Jean-Daniel Zucker, Edi Prifti: ECGrecover: A Deep Learning Approach for Electrocardiogram Signal Completion. KDD (1) 2025: 2359-2370
>
> [2] Jinho Joo, Gihun Joo, Yeji Kim, Moo-Nyun Jin, Junbeom Park, Hyeonseung Im: Twelve-Lead ECG Reconstruction from Single-Lead Signals Using Generative Adversarial Networks. MICCAI (7) 2023: 184-194
>
> [3] Juan Miguel Lopez Alcaraz, Nils Strodthoff: Diffusion-based Time Series Imputation and Forecasting with Structured State Space Models. Trans. Mach. Learn. Res. 2023 (2023)
>
>
> ### **Q5. Clarification of “device heterogeneity”**
>
> **Response:**
>
> Thank you for your careful insight. *Device heterogeneity* refers to systematic differences introduced by the **ECG acquisition hardware** rather than merely differences between datasets. ECG monitors from different manufacturers employ distinct **circuit designs**, **amplification pipelines**, **analog/digital filters**, and **electrode interfaces**, all of which cause device-specific waveform characteristics even for the same patient.
>
> The five datasets used in our study were collected using **five different ECG devices**, summarized below:
>
> | **Dataset** | **Acquisition Device** |
> | ----------- | ---------------------- |
> | Panobench   | g.tec                  |
> | CPSC2018    | Holter                 |
> | Tianchi     | Mindray BeneVision     |
> | PTBXL       | CardioVit CS-12        |
> | ChinaDB     | GE MUSE ECG System     |
>
> Our device calibration step is designed to reduce these device-induced discrepancies. Although the model retains reasonable cross-device generalization without calibration, the goal of our work is to **synthesize clinically meaningful ECG signals** in some clinical settings, where subtle morphological fidelity (e.g., ST-segment shape, repolarization patterns) is crucial for diagnosis.
>
> As shown in **Table 5** and **Figure 5**, device calibration improves the preservation of these physiological details, resulting in more realistic and clinically interpretable synthesized signals.

---

> ### Author Response · Authors · 2025-11-19
> **To Reviewer MVub (Part IV)**
>
> ### **Q6. Medical Oversight in Panobench Collection**
>
> **Response:**
>
> Thank you for your careful insight. Panobench was collected **under direct medical supervision**. Certified clinicians guided electrode placement, verified acquisition protocols, and ensured physiological correctness throughout recording. All sessions followed standard clinical ECG acquisition procedures.
>
> ### **Q7. Preprocessing and Sampling-Rate Normalization**
>
> **Response:**
>
> Thank you for the helpful suggestion. The preprocessing applied to the datasets involved upsampling or downsampling and normalization to the (0, 1) range. Because 500 Hz is the most commonly used clinical sampling rate, all ECG signals were resampled to 500 Hz, and the results reported in the paper correspond to this setting. We observed no notable differences under other sampling configurations in pre-experiments. We have included these preprocessing details in the revised manuscript. Thank you!
>
> ### **Q8. Experimental Setting — Heartbeat-Level vs. Continuous ECG**
>
> **Response:**
>
> All experiments were conducted under the **arbitrary-length** reconstruction and synthesis setting, as mentioned in the Introduction Section and the Methodology Section. As discussed in the Introduction, one limitation of Nef-Net is that it models ECG at the single-heartbeat level. This design requires waiting for each new heartbeat to appear before synthesis can proceed, which introduces substantial latency in real-world applications. Nef-Net+ is explicitly designed to overcome this limitation.
> In addition, although Nef-Net exhibits degraded performance when (extended to) arbitrary-length inputs, we note that even when comparing Nef-Net’s best heartbeat-level results to Nef-Net+ operating on arbitrary-length signals, Nef-Net+ still achieves superior performance, while using only half the number of parameters.
>
> ### **Q9. Implementation Details of Arbitrary-Length Synthesis**
>
> **Response:**
>
> Our method adopts a streamed synthesis strategy, and we will highlight this in the final version. It leverages the spatial relationships between ECG viewpoints to perform view-to-view transformation. Because the model relies on inter-view dependencies rather than constructing a neural electrocardio field as in the original Nef-Net, it can generate signals of arbitrary length by progressively applying these spatially informed transformations across the entire sequence. In practice, as new ECG segments arrive, Nef-Net+ can immediately synthesize the corresponding segments of the target view without waiting for a complete heartbeat cycle, which improves real-time performance.

---

### Official Review · Reviewer_xHLx · 2025-10-31

**Soundness:** 3
**Presentation:** 4
**Contribution:** 3
**Rating:** 6
**Confidence:** 3

**Summary:**

The paper presents Nef-Net+, an improved version of Nef-Net for synthesizing panoramic ECG signals from arbitrary viewpoints, addressing limitations like single-heartbeat modeling, device variability, and electrode placement errors through a new geometry-aware architecture and a three-stage workflow (pretraining, device calibration, on-the-fly calibration).
It also introduces Panobench, a new 48-view ECG dataset with CT-derived coordinates for benchmarking. While the enhancements show promising improvements in signal quality metrics (PSNR and SSIM), the evaluation lacks clinical validation, broader comparisons, and deeper analysis of the method's physiological fidelity, raising questions about its real-world diagnostic utility.

**Strengths:**

- The study is already established on a sound baseline work, and the treatment and the introduction of a geometry-aware cross-attention mechanism (GeoVT) and query-guided feature encoding are shown to be a reasonable solution, effectively addressing the limitations
of uniform feature averaging in prior work, leading to better handling of sparse views and improved synthesis quality.

- The three-stage workflow is a practical, enabling adaptation to device heterogeneity and patient-specific variations, which are critical for clinical deployment.

- Panobench represents a valuable new benchmark with dense 48-view recordings and precise angular annotations, filling a gap in existing datasets and facilitating more comprehensive evaluation of panoramic ECG methods.

- The visual schematics are highly informative and effectively showcases the methodology with clear annotations.

**Weaknesses:**

o The paper claims robustness to "in-the-wild" challenges but tests primarily on controlled datasets; real-world noise (e.g., motion  artifacts, baseline wander) is not explicitly simulated or evaluated beyond angular deviations.

o The evaluation relies heavily on PSNR and SSIM, which measure signal similarity but do not assess clinical relevance, such as the preservation of diagnostic features (e.g., arrhythmia detection or pathology-specific waveforms). Other potential metrics, like sensitivity/specificity for disease classification on synthesised signals, could strengthen claims of clinical utility.

o Comparisons are limited to Nef-Net and a few others (KIM and E-LSTM); the paper does not benchmark against other recent ECG synthesis methods (e.g., GAN-based or transformer-based approaches cited in related work), potentially overstating the novelty and superiority of NEF-NET+.

o Panobench has limited demographic diversity (subjects aged 18-28), which may not capture variations in older populations or those with comorbidities, limiting generalizability. Additionally, the dataset size (9360 recordings) is substantial but lacks details on
pathological distribution beyond basic categories.

o The on-the-fly calibration uses only the first 5 seconds of a 10-second recording, assuming stability; however, this may not hold for arrhythmic or dynamic ECGs, and no sensitivity analysis is provided for calibration duration or failure cases. This may highlight potential
ablation studies in this direction.

**Questions:**

o Can the authors elaborate on the rationale for employing the dipole approximation from cardiac vector theory as the foundational model, and discuss how NEF-NET+ accounts for potential higher-order multipolar contributions in scenarios involving complex cardiac
pathologies?

o Regarding the GeoVT module, the utilization of a shared Geometric Angular Attention (GAA) map across all blocks implies an assumption of static angular similarity; could this constrain the model's capacity to handle temporal variations in cardiac electrical activity, and were alternative dynamic attention mechanisms explored during development?

o Please provide details on the initialization and optimization strategy for the angular deviation parameters (dθ, dφ) in the on-the-fly calibration phase, including measures implemented to mitigate overfitting to the initial 5-second segment under conditions of signal noise or variability.

o In Table 4, the observed performance improvements vary across disease categories (e.g., relatively modest gains for LBBB); what factors might contribute to these disparities, and do they suggest inherent limitations in the model's representation of specific conduction
disorders?

o Could the authors explain the absence of assessments for downstream applications, such as leveraging synthesized ECG signals to enhance diagnostic model training or conducting clinician-led perceptual fidelity studies?

o Appendix E.2 demonstrates the efficacy of calibration for angular deviations up to 30°; what is the documented range of electrode placement errors in clinical practice, and how does the model's performance scale for deviations exceeding this threshold?

---

> ### Author Response · Authors · 2025-11-19
> **To Reviewer xHLx (Part I)**
>
> Thank you for your positive evaluation of our work, including the effectiveness of the geometry-aware GeoVT and query-guided encoding, the practicality of the three-stage workflow for clinical deployment, the value of PanoBench in addressing the gap in multi-view ECG datasets, and the informativeness of our visual schematics. Below we provide a point-by-point response to your questions and hope it clarifies all of your concerns.
>
> ### **Q1: The paper claims robustness to "in-the-wild" challenges but tests primarily on controlled datasets; real-world noise (e.g., motion artifacts, baseline wander) is not explicitly simulated or evaluated beyond angular deviations.**
>
> **Response:**
>
> **(1) Clarification of “in-the-wild”**
>
> In our paper, “in-the-wild” denotes ECG recordings collected from human subjects under real-world conditions. The central challenge in such settings arises from device variation and deviations in electrode placement, as outlined in the Introduction. These issues remain unaddressed in the original Nef-Net. Our approach is designed to correct these real-world variations through Device Calibration and On-the-fly Calibration respectively. The five public datasets used in this study were collected in the wild with five different devices (as listed below) in non-controlled environments, which fully meets the definition of “in-the-wild”.
>
> | **Dataset** | **Acquisition Device** |
> | --- | -- |
> | Panobench   | g.tec                  |
> | CPSC2018    | Holter                 |
> | Tianchi     | Mindray BeneVision     |
> | PTBXL       | CardioVit CS-12        |
> | ChinaDB     | GE MUSE ECG System     |
>
>
> **(2) Evaluation on noisy data**
>
> Nef-Net+ is intrinsically robust to common ECG distortions such as **powerline interference** and **baseline wander**, because these artifacts are largely low-frequency or harmonic distortions that are attenuated during the generative process. Motion artifacts, however, constitute a fundamentally different category of disturbance. ECGs corrupted by motion artifacts should be **discarded rather than reconstructed** as discussed in both clinical workflows and biomedical engineering literature, since artifact suppression depends primarily on acquisition-level hardware design rather than post-hoc generative modeling [1–2].
>
> To further assess robustness to common noise in a quantitative manner, we conducted two analyses across the four datasets, excluding Panobench (as Panobench is more clean):
>
> - **Testing with noisy reference signals**
>
> We identified samples affected by **powerline interference** or **baseline drift** and compared the synthesis quality before and after denoising.
>
> | Noise Type | ChinaDB (PSNR/SSIM) | CPSC2018 (PSNR/SSIM) | Tianchi (PSNR/SSIM) | PTBXL (PSNR/SSIM) |
> |----|-----|--------|---|----|
> | Powerline interference | 32.14 / 0.979 | 33.11 / 0.983 | 34.57 / 0.980 | 33.04 / 0.981 |
> | Baseline wander | 32.53 / 0.982 | 33.47 / 0.983 | 34.69 / 0.981 | 33.25 / 0.981 |
> | Clear signal | 33.24 / 0.986 | 34.18 / 0.987 | 35.14 / 0.987 | 34.17 / 0.983 |
>
> **In summary, noise exerts only a limited influence on our model.** Notably, because the generator tends to produce “clean” signals, discrepancies appear when the reference contains noise. This lowers the raw metrics even when the generated output is visually superior, which means the actual impact may be even smaller than the table suggests. We have included these results in the revised version.
>
> - **Testing after filtering reference signals**
>
> We therefore repeated the evaluation after denoising the reference signals to ensure a fair comparison.
>
> | Noise Type | ChinaDB (PSNR/SSIM) | CPSC2018 (PSNR/SSIM) | Tianchi (PSNR/SSIM) | PTBXL (PSNR/SSIM) |
> |---|--|----|---|---|
> | Powerline interference | 33.21 / 0.985 | 34.09 / 0.985 | 35.12 / 0.987 | 34.04 / 0.982 |
> | Baseline wander | 33.08 / 0.984 | 34.13 / 0.985 | 34.97 / 0.986 | 33.95 / 0.982 |
> | Clear signal | 33.24 / 0.986 | 34.18 / 0.987 | 35.14 / 0.987 | 34.17 / 0.983 |
>
> Across all datasets, the performance remains stable after denoising, demonstrating that **Nef-Net+ is robust to common ECG noise** and tends to remove such artifacts during generation. We appreciate the reviewer’s insight, and we have included these results in the revised version.
>
> **(3) Evaluated beyond angular deviations**
>
> At this point, we have incorporated the discussion into our response to Q10. Please refer to that section. Thank you!
>
> [1] Torfs T, Chen Y H, Kim H, et al. Noncontact ECG recording system with real time capacitance measurement for motion artifact reduction[J]. IEEE transactions on biomedical circuits and systems, 2014.
>
> [2] Lee S Y, Su P H, Hung Y W, et al. Motion artifact reduction algorithm for wearable electrocardiogram monitoring systems[J]. IEEE Transactions on Consumer Electronics, 2023.

---

> ### Author Response · Authors · 2025-11-19
> **To Reviewer xHLx (Part II)**
>
> ### **Q2. The evaluation relies heavily on PSNR and SSIM; do not assess clinical relevance, such as the preservation of diagnostic features.**
>
> **Response:**
>
> To assess whether our method preserves pathological characteristics, the experiment in Section 4.4 has demonstrated that Nef-Net+ synthesizes morphology-consistent signals in abnormal cases. Moreover, Figure 5 shows that our three-stage development pipeline reconstructs clinically meaningful features. For instance, in samples 2300–2800, the synthesized T-wave amplitude decreases below the R-wave, consistent with the true diagnostic pattern.
>
> We will further strengthen this validation with two additional experiments:
>
> - View-specific pathological reconstruction: We include synthesized panoramic signals for left and right bundle branch block, both of which exhibit distinct and view-dependent waveforms. The results show that these characteristic conduction patterns are preserved.
>
> - Downstream diagnostic evaluation and Cardiologist review: We train an 1D classifier (following reference [1]) as a proxy for clinical assessment on some common heart diseases (AF, LBBB, RBBB, PAC) and consult two cardiologists to evaluate the pathological validity of our generated signals on 30 cases.
>
> #### **Diagnostic classifier performance**
> |      | AF   | LBBB | RBBB | PAC
> |-----|--|--|--|------|
> | Three-leads       | 0.81 | 0.71 | 0.72 | 0.58
> | Generated 12-lead | 0.82 | 0.77 | 0.79 | 0.68
> | Real 12-lead   | 0.84 | 0.78 | 0.83 | 0.71
>
> **Our results show that the model captures diseased patterns with high fidelity**
>
> The synthesized signals for LBBB and RBBB conditions in key leads (V1 and V6) were evaluated by two independent expert cardiologists. The diagnostic quality of characteristic waveforms was rated on a 0-5 scale (5 is better). All scores fell within the range of 3 to 5, indicating clinically acceptable to excellent feature fidelity.
>
> #### **Cardiologist 1 evaluation**
> | Diagnosis | 5 | 4 | 3| 2 | 1 | 0 | Total |
> |--|---|--|---|---|---|---|---|
> | LBBB      | 25  | 5  | 0 | 0  |0 |0  | 30  |
> | RBBB      | 28 | 2  | 0  |0  |0 |0 | 30  |
>
> #### **Cardiologist 2 evaluation**
> | Diagnosis | 5 | 4 | 3 | 2 | 1 | 0 | Total |
> |--|---|--|---|---|---|---|---|
> | LBBB      | 25          | 3    | 2     | 0 | 0 | 0  | 30    |
> | RBBB      | 29          | 0    | 1   | 0 | 0 | 0      | 30   |
>
> These evaluations collectively show that our model not only preserves high structural similarity but also maintains *diagnostic integrity* across multiple abnormal conditions. We have included these results in the revised version.
>
> [1] Jie Hu, Li Shen, Gang Sun, Squeeze-and-excitation networks, CVPR 2018.
>
> ### **Q3. Limited Comparison Against ECG Reconstruction Methods**
>
> **Response:**
>
> We would like to clarify that the reconstruction experiment in our paper serves only as a **baseline to contextualize synthesis performance**, as stated in the Experiment Section ("...reconstruction performance serves as an upper bound..."). For this reason, our original submission did not compare extensively with reconstruction-oriented models.
>
> To address this concern, we expanded our benchmarking. All models were reproduced under the **3-to-9** setting using the same data splits and preprocessing for fair comparison.
>
> #### **Reconstruction performance comparison**
>
> | Method      | ChinaDB (PSNR/SSIM) | CPSC2018 (PSNR/SSIM) | Tianchi (PSNR/SSIM) | PTB-XL (PSNR/SSIM) |
> |-------------|-----|------------------------|----------------------|----------------------|
> | ECGRecover [1]  | 30.47 / 0.958        | 30.12 / 0.966          | 31.47 / 0.971        | 28.57 / 0.942        |
> | EKGAN [2]    | 32.76 / 0.967        | 33.35 / 0.975          | 34.39 / 0.977        | 31.71 / 0.972        |
> | SSSD [3]    | 32.53/0.966    |32.67/0.972         | 33.59/0.975             | 31.42/0.972                   |
> | **Nef-Net**     | 29.59 / 0.961        | 29.12 / 0.958          | 31.44 / 0.965        | 30.22 / 0.962        |
> | **Nef-Net+** | **35.84 / 0.977** | **36.12 / 0.981**       | **37.13 / 0.982**     | **35.21 / 0.974**     |
>
> These results show that while our framework is not designed specifically for lead-reconstruction, **Nef-Net+ achieves substantially stronger performance** than all reconstruction-focused baselines across multiple datasets.
> We appreciate the reviewer’s insight, and we have included these results in the revised version.
>
> [1] Alex Lence, Federica Granese, Ahmad Fall, Blaise Hanczar, Joe-Elie Salem, Jean-Daniel Zucker, Edi Prifti: ECGrecover: A Deep Learning Approach for Electrocardiogram Signal Completion. KDD (1) 2025: 2359-2370
>
> [2] Jinho Joo, Gihun Joo, Yeji Kim, Moo-Nyun Jin, Junbeom Park, Hyeonseung Im: Twelve-Lead ECG Reconstruction from Single-Lead Signals Using Generative Adversarial Networks. MICCAI (7) 2023: 184-194
>
> [3] Juan Miguel Lopez Alcaraz, Nils Strodthoff: Diffusion-based Time Series Imputation and Forecasting with Structured State Space Models. Trans. Mach. Learn. Res. 2023 (2023)

---

> ### Author Response · Authors · 2025-11-19
> **To Reviewer xHLx (Part III)**
>
> ### **Q4. Limited demographic diversity and lack of detailed pathological distribution in Panobench**
>
> **Response:**
>
> Panobench was collected using a **high-density 48-view ECG acquisition system** (~USD 210,000). The device is **large and heavy**, and each recording requires **close to one hour** to position and attach 46 electrodes, which is very uncomfortable for subjects. Due to the equipment cost, operational complexity, and long setup time, **recruitment of patients—especially those with comorbidities—is extremely difficult**. This practical constraint explains why Panobench primarily contains healthy young adults.
>
> Given these limitations, our evaluation adopts a **deliberate two-stage strategy**:
>
> **(1)** Panobench is **only** used to evaluate the core contribution of our work—
>    **full-view panoramic synthesis and geometric consistency**, which require dense multi-angle recordings that only Panobench provides.
>
> **(2)** Clinical generalization and pathology-related performance are evaluated on **large, pathologically diverse public datasets** (e.g., CPSC2018, Tianchi, PTBXL), as shown in Table 2&5.
>
> This decoupled evaluation design is a **necessary and appropriate compromise** as Panobench enables geometric testing and standard benchmarks enable pathological validation.
>
> Despite the challenges, we are continuing patient recruitment. Over the past three years, we have already collected recordings from 7 heart disease patients using this 48-view system. Although this number is not yet sufficient to form a full dataset, we intend to expand it in future work and conduct a dedicated patient-specific evaluation.
>
> ### **Q5. Sensitivity of On-the-Fly Calibration to Calibration Duration**
>
> **Response:**
>
> The on-the-fly calibration module is designed to correct **view-angle (spatial) deviations**, not temporal dynamics. Since lead-angle geometry is a spatial property of the cardiac electrical field, it remains stable regardless of arrhythmia or rhythm variability. Thus, calibration does not rely on long-term temporal consistency. Our choice of a **5-second calibration window** is supported by the following:
>
> - Clinical feasibility. Standard ECG acquisition uses 10-second segments. We discussed with some cardiologists and confirmed that extending the calibration window (e.g., to ~15 seconds) is clinically feasible if needed.
>
> - Recording stability. Clinical ECGs are collected with the patient lying still, spatial deviations (e.g., electrode angle offsets) are stable during acquisition, making short windows sufficient for estimating them.
>
> - Sensitivity analysis. We conducted an ablation study using 3–7 second calibration segments. Performance improves rapidly from 3 to 5 seconds, with diminishing returns thereafter, indicating that **5 seconds is a near-optimal trade-off**.
>
> | Calibration Duration | Before | After  | Δ (Improvement) |
> |----------------------|--------|--------|------------------|
> | 3s                   | 32.08  | 33.18  | -                |
> | 4s                   | 32.08  | 33.45  | 0.27             |
> | **5s**               | **32.08** | **33.62** | **0.17**         |
> | 6s                   | 32.08  | 33.71  | 0.09             |
> | 7s                   | 32.08  | 33.76  | 0.05             |
>
> ### **Q6. Rationale for Using the Dipole Approximation and Handling Higher-Order Effects**
>
> **Response:**
>
> The dipole approximation is the classical formulation of cardiac vector theory and models cardiac activity as a **latent 3D electrical vector**, whose projections form the observed ECG leads (lines 727–745). Even under pathological conditions, deviations in morphology can be interpreted as **altered realizations of this latent vector**, which remains a compact and physiologically meaningful representation.
>
> Nef-Net+ does not explicitly reconstruct a full “neural electrocardio field” or estimate higher-order multipolar components (lines 703–726). Approaches that attempt multipole-based modeling face two major challenges:
>
> - Parameter explosion: higher-order expansions require many coefficients, making them difficult to learn and computationally impractical for high-density or arbitrary-view synthesis.
> - Poor generalization: pathological conditions substantially perturb the cardiac field, making explicit multipolar modeling fragile and inconsistent across diverse disease states.
>
> Our method instead adopts a geometry-aware view-transformation strategy, which avoids explicit multipole reconstruction. Nef-Net+ determines whether the available input views carry sufficient morphological information for the target view, and synthesizes the waveform by **aggregating features through learned inter-view spatial relationships**. This allows the model to implicitly capture complex morphology—including higher-order effects—without relying on unstable explicit multipole estimation.

---

> ### Author Response · Authors · 2025-11-19
> **To Reviewer xHLx (Part IV)**
>
> ### **Q7. Static GAA Map & Potential Need for Dynamic Attention**
>
> **Response:**
>
> The Geometric Angular Attention (GAA) encodes **only the spatial layout of electrodes**, which is fixed and does not depend on the temporal behavior of cardiac activity. Since the angular relationships between leads are entirely determined by electrode placement, a **static angular similarity map is the correct formulation** and does not constrain the model’s ability to capture temporal variations.
>
> We have explored several dynamic variants during the project, including:
>
> - Feature-conditioned GAA, where angular weights were updated jointly using signal features and positions.
> - Time-varying angular attention, where angle-based weights were dynamically recomputed across temporal segments.
> - Hybrid dynamic-static formulations incorporating both geometric priors and feature-driven updates.
>
> None of these alternatives provided measurable improvements, and in several cases, they introduced instability due to unnecessary fluctuations in the geometric prior.
>
> ### **Q8. Initialization and Optimization of Angular Deviations (dθ, dφ) and Measures Against Overfitting**
>
> **Response:**
>
> **(1)** Details of the optimization setup are provided in Appendix D. During the **on-the-fly calibration** stage, only the angular deviation parameters (dθ, dφ) are updated, while the view encoder and synthesis modules remain frozen. The parameters (dθ, dφ) are **initialized to zero**, optimized for 100 iterations, with a learning rate of 5×10⁻⁵.
>
> **(2)** The risk of overfitting during on-the-fly calibration is inherently low. The learning rate is set to a very small value, the calibration window spans only five seconds, and a substantial portion of the model remains frozen during this stage. These constraints effectively limit the degree of optimization and prevent overfitting. An ablation study provided in our response to Q5 further shows that our calibration setting does not lead to overfitting.
>
> ### **Q9. Uneven Performance Gains Across Disease Categories (e.g., LBBB)**
>
> **Response:**
>
> Actually, the relatively modest improvement for LBBB in Table 4 is primarily due to data quality issues in the CPSC2018 LBBB subset. Several recordings show partial lead detachment, heavy motion artifacts, or abrupt large-amplitude spikes—samples that would normally be discarded in clinical practice. To ensure comparability with future work, we reported results on the **original unfiltered dataset**.
>
> | Dataset Setting | PSNR | SSIM |
> |-----------------|-------|-------|
> | All LBBB samples | 28.35 | 0.955 |
> | After removing low-quality samples | **31.77** | **0.975** |
>
> When we remove contaminated LBBB segments, synthesis performance improves substantially. The results show that the apparent disparity is **data-driven**, not a limitation of the model.
>
> In terms of representational capacity, the **geometry-aware view-transformation mechanism** is not tied to any specific pathology. Given informative input views, it can reconstruct the distinctive morphological patterns of conduction abnormalities, such as prolonged QRS complexes or altered ST–T morphology, without requiring a pathology-specific architectural design. This is a key improvement over Nef-Net, which is more sensitive to variations in pathological electrical fields.
>
> ### **Q10. Electrode Placement Error Range and Model Behavior Beyond 30° Deviation**
>
> **Response:**
>
> Clinical studies report that electrode placement deviations typically fall within **5°–20°**, depending on technician experience and patient posture. In our dataset, Panobench exhibits an **average angular standard deviation of 10.6°**, with the **maximum observed deviation at 25.63°** (as stated in Sec. 3.2).
>
> Appendix E.2 shows that model performance degrades progressively as deviations increase. Each additional **10°** of angular error produces approximately a **1–2 dB PSNR drop** and a **0.04–0.15 SSIM drop** if uncorrected.
>
> The calibration module restores performance well up to **~30°**, but beyond this range, synthesized signals begin to lose physiologically meaningful structure, which is consistent with clinical expectations, as deviations above 30° correspond to substantially misplaced electrodes rather than realistic acquisition noise.
>
> | Deviation | Uncorrected (PSNR/SSIM) | After Calibration (PSNR/SSIM) |
> |---|--|-----|
> | 10° | 30.71 / 0.971 | 33.24 / 0.983 |
> | 20° | 28.79 / 0.965 | 33.09 / 0.982 |
> | 30° | 26.53 / 0.960 | 33.09 / 0.982 |

---

### Author Response · Authors · 2025-11-28
**Summary (Part I)**

We thank all reviewers for their thoughtful evaluations and for recognizing the contribution of our work. Their insights have been invaluable, and incorporating their suggestions has substantially strengthened the manuscript. We are grateful that the reviewers found our contributions meaningful. In particular:

1. We appreciate the recognition of the **clarity and overall quality of our presentation** (*R1*).

2. We thank the reviewers for noting that our **novel architecture effectively addresses the limitations of earlier approaches confined to controlled experimental settings**, achieving higher synthesis fidelity (*R1*, *R2*, *R3*).

3. We value the acknowledgment that our three-stage calibration framework **reflects real-world ECG acquisition workflows** and is essential for practical clinical deployment (*R1*, *R3*).

4. We are also grateful for the recognition of **the value of our new Panobench dataset** (*R1*, *R2*, *R3*).

For clarity, we restate our primary contributions:

1. **Advancing Electrocardio Panorama from laboratory paradigms to real-world practice:** We introduce a new view-to-view paradigm with a new model architecture that explicitly models the spatial relationships between the query view and recorded ECG views. This enables the model to identify only the most relevant features for direct query-view transformation. The formulation resolves the feature-averaging limitation in previous work Nef-Net and allows NEF-NET+ to achieve markedly improved synthesis performance (for example, 32.07 (+7.97 compared to previous work) to 34.82 (+6.81 compared to previous work) in PSNR while reducing model complexity. This improvement in adaptability has clear potential for patient benefit in real clinical workflows.

|Method|ECGLength|Device|LeadPlacement|SynthesisPSNR↑
|-|-|-|-|-
|Nef-Net|Heartbeat|Restricted|High-Precision|24.10–28.01
|**NEF-NET+**|Continuous|Agnostic|In-the-wild|**32.07(+7.97)**–**34.82(+6.81)**

2. **A scenario-driven and effective three-stage development pipeline:** We propose a three-stage pipeline that includes large-scale pretraining, device-specific calibration for hospitals and rapid on-the-fly calibration for individual patients. These stages directly address two **real-world** persistent clinical challenges: variability across ECG devices and inconsistency in operator technique.

3. **A high-value benchmark for the community:** We present Panobench, a dataset comprising **9,360 ECG recordings** collected using clinical-grade equipment costing approximately **USD 210,000**, with each recording requiring nearly **1 hour of patient preparation** and acquisition. It is **the first 48-lead ECG dataset** and, for the first time, provides CT-derived spherical coordinates (θ, φ). We believe this dataset offers substantial value not only for Electrocardio Panorama synthesis but also for broader cardiac electrophysiology research.

Beyond these strengths, the reviewers offered constructive suggestions that have further improved the manuscript. In response:

1. We have uploaded an updated PDF with all revisions shown in blue, including expanded analyses of pathological patterns in Figures 10–13 and Appendix E.5, as requested by the reviewers.

2. We have provided detailed, line-by-line responses to all reviewer comments and trust that they address every concern (**as summarized in Part II & III**).

3. We further summarized the key points in Parts II and III to facilitate clearer review.

---

> ### Author Response · Authors · 2025-11-29
> **Summary (Part II)**
>
> **1. Clinical Validity of Synthesized Signal Morphology**
>
> The original submission has partly evaluated the ability of NEF-NET+ to synthesize morphology-consistent signals in abnormal cases in Section 4.4. Figure 5 also illustrates waveform characteristics across development stages and provides a qualitative analysis. To further assess the clinical validity of NEF-NET+, the additional results are included in the revised version:
>
> (1) Pathology-specific morphological analysis of synthesized signals for LBBB and RBBB in their clinically critical views (V1 and V6) as example in Fig.10-11.
>
> (2) Downstream validation shows that the synthesized 12-lead ECGs enable clinical classification performance that closely matches that obtained with real 12-lead ECGs, and they are better than the model trained on the 3-lead recorded signals.
>
> #### **Diagnostic classifier performance**
> ||AF|LBBB|RBBB|PAC
> |-|-|-|-|-|
> |3-leads|0.81|0.71|0.72|0.58
> |Generated12-lead|0.82|0.77|0.79|0.68
> |Real12-lead|0.84|0.78|0.83|0.71
>
> (3) We invited two cardiologists to evaluate 30 synthesized diseased ECGs, scoring 0-5 according to whether the pathological morphology was discernible (higher is better, 3 is qualified). The results show that NEF-NET+ generates the morphological cues required for clinical judgment.
>
> #### **Cardiologist 1**
> ||5|4|3|2|1|0|Total
> |-|-|-|-|-|-|-|-|
> |LBBB|25|5|0|0|0|0|30
> |RBBB|28|2|0|0|0|0|30
>
> #### **Cardiologist 2**
> ||5|4|3|2|1|0|Total
> |-|-|-|-|-|-|-|-|
> |LBBB|25|3|2|0|0|0|30
> |RBBB|29|0|1|0|0|0|30
>
> (4) We include synthesized 48-view visualizations, compared with ground-truth on PanoBench (Fig.12-13). The synthesized and real views are closely aligned, indicating that NEF-NET+ captures morphological structure with remarkable fidelity.
>
> **2. Three-stage development pipeline and deployment considerations**
>
> The three-stage framework comprises pre-training on a large-scale dataset, device-specific calibration, and on-the-fly calibration, which are designed to learn general representations, adapt to specific devices, and perform real-time adjustment for   real-world operation deviation, respectively.
>
> - *(MVub) What device heterogeneity refers to?*
>
> Device heterogeneity refers to systematic differences introduced by the ECG acquisition hardware. ECG monitors from different manufacturers employ distinct circuit designs, amplification pipelines, analog/digital filters, all of which cause device-specific waveform characteristics even for the same patient. In the experiments, the 5 datasets used were collected with 5 different devices.
>
> - *(yWqz) The feasibility of performing on-the-fly calibration.*
>
> NEF-NET+ is a lightweight view-transformation model with 3.8M parameters. By avoiding explicit electric-field reconstruction, in contrast to Nef-Net, and performing only geometry-aware view transformation, the on-the-fly calibration stage updates fewer than 6% of the parameters, which are solely angle related. Considering the performance–efficiency balance and feedback from collaborating clinicians, **a calibration window of 5 seconds was identified as optimal**. The on-the-fly calibration over 100 iterations using an NVIDIA GeForce RTX 2080 Ti GPU requires approximately 6 seconds (Appendix D.3), and this computation can overlap with the 5-second recording process. These results indicate that on-the-fly calibration is practical even for resource-constrained medical hardware.

---

> ### Author Response · Authors · 2025-11-29
> **Summary (Part III)**
>
> **3. A few Technical Clarifications of Nef-Net & NEF-NET+**
>
> - *(xHLx) The Theory Behind NEF-NET+ and Nef-Net.*
>
> While Nef-Net synthesizes novel ECG views by modeling the underlying ECG field, NEF-NET+ simplifies this process by a direct view transformation according to cardiac vector theory (explained in Appendix B). This is achieved by leveraging the geometric relationships between target views and recorded views. Specifically, it employs a Geometry-Aware Attention mechanism to weight features from the recorded views that are most relevant to the target views, enabling effective synthesis.
>
> - *(yWqz) Limited Validation of Geometry-Aware Attention's function.*
>
> To address this concern, we provide quantitative evidence that the Geometry-Aware Attention and GeoVT modules indeed learn anatomically meaningful inter-lead relationships.
>
> #### **(1) Inputs: Lead I, Lead II, V1 → Generating V2–V6**
> ||V2|V3|V4|V5|V6|
> |-|-|-|-|-|-|
> |Lead-I|0.21|0.37|0.36|0.35|0.35|
> |Lead-II|0.15|0.29|0.33|0.37|0.43|
> |Lead-V1|0.64|0.34|0.28|0.28|0.22|
>
> #### **(2) Inputs: Lead I, Lead II, V3 → Generating V1, V2, V4, V5, V6**
> ||V1|V2|V4|V5|V6|
> |-|-|-|-|-|-|
> |LeadI|0.48|0.29|0.25|0.28|0.33|
> |LeadII|0.12|0.15|0.15|0.29|0.36|
> |V3|0.40|0.55|0.60|0.43|0.31|
>
> #### **(3) Inputs: Lead I, Lead II, V5 → Generating V1–V4, V6**
> ||V1|V2|V3|V4|V6|
> |--------|------|------|------|------|------|
> |LeadI|0.41|0.37|0.35|0.14|0.13|
> |LeadII|0.29|0.29|0.19|0.13|0.12|
> |V5|0.30|0.34|0.46|0.73|0.75|
>
> One can see that the attention weights exhibit a strong positive correlation with the geometric proximity between the synthesized and target views. This behavior aligns exactly with the known physical chest-lead geometry.
>
> We further computed pairwise Pearson correlations among chest leads based on their corresponding ECG signals:
> ||V1|V2|V3|V4|V5|V6|
> |-|-|-|-|-|-|-|
> |V1|1|0.87|0.45|0.15|0.11|0.11|
> |V2|0.87|1|0.76|0.43|0.39|0.38|
> |V3|0.45|0.76|1|0.88|0.84|0.83
> |V4|0.15|0.43|0.88|1|0.99|0.98
> |V5|0.11|0.39|0.84|0.99|1|0.99
> |V6|0.11|0.38|0.83|0.98|0.99|1
>
> The results show that the learned attention patterns follow the actual geometric relationships:
> 1. The model assigns highest attention to the anatomically closest available lead.
> 2. Attention weights decay smoothly with increasing inter-lead anatomical distance.
> 3. The pattern is consistent whether the mid-chest (V3) or lateral leads (V5) are provided as inputs.

---

### Meta-Review · Area_Chair_kfRk · 2026-01-06

**Summary:**

The paper presents Nef-Net+, a framework for panoramic ECG synthesis that improves upon the prev Nef-Net by addressing practical "in-the-wild" challenges like device heterogeneity, electrode placement deviations, and arbitrary signal lengths. The authors introduce a geometry-aware view transformation and a three-stage calibration pipeline (pretraining, device calibration, on-the-fly calibration). A significant contrib is "Panobench," a new dataset with 9,369 recordings of 48-view ECGs.

**Reviewer Concerns:**

The discussion phase focused on three main areas:

1. **Scope and Relevance (Reviewer MVub)**: Argued the paper was out of scope for ICLR.
   - *Addressed?*: **Yes (by AC).** I explicitly overrule this concern. The paper fits well within the "Applications to Healthcare" track. The contribution is both methodological (geometry-aware attention) and resource-based (dataset).

2. **Clinical Validation (Reviewers xHLx, yWqz)**: Noted that standard metrics (SSIM/PSNR) are insufficient.
   - *Addressed?*: **Yes.** The authors conducted a new cardiologist evaluation study (2 experts, 30 cases) and a downstream diagnostic classification experiment. Although reviewers did not verify this, the quantitative data (e.g., preservation of LBBB/RBBB features) seems convincing.

3. **Methodological Details**: Concerns about the "black box" nature of the attention mechanism.
   - *Addressed?*: **Yes.** The authors provided correlation analysis showing attention weights match anatomical distances.

**Reviewer Scores:**

- Reviewer MVub: 2
- Reviewer xHLx: 6
- Reviewer yWqz: 6

---

### Decision · Program_Chairs · 2026-01-26

Accept (Poster)